# Capillary compression induced outstanding n-type thermoelectric power factor in CNT films towards intelligent temperature controller

Hong Wang [1,2,3] ✉, Kuncai Li[1,3], Xin Hao[1,2], Jiahao Pan[1], Tiantian Zhuang[1], Xu Dai[1], Jing Wang[1,2], Bin Chen [1,2] & Daotong Chong[1,2]

One-dimensional carbon nanotubes are promising candidates for thermoelectrics because of their excellent electrical and mechanical properties. However, the large n-type power factor remains elusive in macroscopic carbon nanotubes films. Herein, we report an outstanding n-type power factor of 6.75 mW m$^{-1}$ K$^{-2}$ for macroscopic carbon nanotubes films with high electrical and thermal conductivity. A high-power density curl-able thermoelectric generator is fabricated with the obtained carbon nanotubes films, which exhibits a high normalized power output density of 2.75 W m$^{-1}$ at a temperature difference of 85 K. The value is higher than that of previously reported flexible all-inorganic thermoelectric generators (<0.3 W m$^{-1}$). An intelligent temperature controller with automated temperature-controlling ability is fabricated by assembling these thermoelectric generators, which demonstrates the potential application of the carbon nanotubes films in automated thermal management of electronic devices where requires a large thermoelectric power factor and a large thermal conductivity simultaneously.

Thermoelectric (TE) materials can realize the direct conversion between heat and electricity, which offers promising potential in the applications of both waste heat recovery and solid-state cooling[1]. For waste heat recovery, previous works typically focus on reducing the thermal conductivity ($k$) to improve the figure-of-merit of TE materials ($ZT$) (which is defined as $ZT = S^2\sigma T/k$, where $S$ is the Seebeck coefficient, $\sigma$ is the electrical conductivity, $T$ is the absolute temperature) to increase the heat-to-electricity conversion efficiency in the application of waste heat recovery[2–4]. While the heat is unlimited and wasted freely in many practical waste heat recovery applications, such as the waste heat in industry and vehicle[5,6]. Recent works suggest that the primary concern is not to pursue high energy efficiency but to generate enough electric power to support the electronic devices in these scenarios[7,8].

Therefore, enhancing the power factor ($PF = S^2\sigma$) to improve the output power density is more important than pursuing high $ZT$ to improve the heat-to-electricity conversion efficiency[7,8]. For solid-state cooling, TE cooling devices enable the heat flowing from the cold side to the hot side through the Peltier effect. It is very attractive for the thermal management of electronic devices e.g., controlling the temperature of CPU and Infrared night-vision camera[9,10], etc. Because the all-solid-state TE cooling devices work quietly with no moving parts and liquid lubricants, which is distinct with traditional refrigeration cooling techniques. The heat flow rate is proportional to the effective thermal conductivity ($k_{eff}$) that is defined as $k_{eff} = k + PF \cdot T_H^2/(2\Delta T)$, where $T_H$ and $\Delta T$ is the temperature of the hot side temperature and the temperature difference between the two sides of the TE device.

[1]State Key Laboratory of Multiphase Flow in Power Engineering & Frontier Institute of Science and Technology, Xi'an Jiaotong University, Xi'an 710054, China. [2]School of Energy and Power Engineering, Xi'an Jiaotong University, Xi'an 710054, China. [3]These authors contributed equally: Hong Wang, Kuncai Li. ✉e-mail: hong.wang@xjtu.edu.cn

It indicates that large $k$ and large $PF$, instead of $ZT$, are highly desired in the applications of solid-state cooling.

Besides the basic TE properties, the rigidity, the toxicity, and the large scalability of TE materials are also key factors for practical applications. However, it is challenging to combine these properties with the high TE properties. On the one hand, inorganic materials have high TE properties, such as $Bi_2Te_3$ exhibiting large $PF$ of ~4.5 mW m$^{-1}$ K$^{-2}$ at room temperature[11], however, which are typically rigid, toxic, and hard to process. On the other hand, organic materials are generally flexible, non-toxic, and easy to process while they have small $PF$, often less than 1 mW m$^{-1}$ K$^{-2}$, especially for n-type organic materials[12,13]. The TE materials, currently being explored, are expected to possess inorganic-like large TE performance and organic-like process ability and environmentally friendly properties.

Recent progress in fabricating flexible carbon nanotube (CNT) macroscopic films[14–17] suggests that CNT films are promising for making the ideal TE materials with inorganic-like large $PF$ and organic-like process ability simultaneously. Low-dimensional (2D and 1D) materials are typically considered to be holding the promise of TE materials due to the narrow carrier distribution derived from the 2D or 1D quantum confinement[18–20]. High $PF$s have been achieved in 2D TE materials, such as 36.6 mW m$^{-1}$ K$^{-2}$ for single layer graphene[21] and 26 mW m$^{-1}$ K$^{-2}$ for ultra-thin FeSe[22]. Theoretically, 1D materials such as CNTs even have a higher $PF$ of 100 mW m$^{-1}$ K$^{-2}$ [23].

However, the $PF$ of CNT films is small compared to that of inorganic materials, especially for n-type materials[24]. To improve the $PF$ of CNT films, separating semiconducting CNTs is an effective method because semiconducting CNTs possess high $S$[25,26]. While the complex separation process requires to cut CNT into small pieces and introduce non-conductive additives, which scarifies the $\sigma$ too much, subsequently limiting the improvement of $PF$[27]. An alternative strategy to improve the $PF$ of CNT films is to tune the Fermi energy ($E_f$) close to a 1D van Hove singularity of CNT. It may simultaneously improve the $\sigma$ and $S$ of CNTs due to the narrow carrier distribution[18] and discontinuous band structure of CNTs derived from the 1D quantum confinement[19,20]. Successful example have been reported experimentally with a high p-type $PF$ up to $14 \pm 5$ mW m$^{-1}$ K$^{-2}$ because of the careful controlling of doping level to tune the $E_f$ as well as the good alignment of CNTs[14,28]. However, n-type high $PF$s of CNT films are still lacking. The $PF$s of CNT are behind that of n-type inorganic materials. Although the commonly used solution process method of mixing CNT powder and the n-type dopants in solvents can provide sufficient n-type doping, the CNTs are typically randomly packed in the filtrated films[29]. It is still challenging to have a material process to make highly ordered CNT assemblies in the macroscopic CNT films together with n-type dopants for suitable n-type doping.

The film TE generators (TEGs) are often used for the recovery of waste heat at low to medium temperatures[2,15,30,31], which have great potential for application in the field of wearable electronic devices due to their good flexibility as compared with pellet structure TEGs. The thin-film TEGs usually prepared from organic TE materials also have the advantages of being lightweight, inexpensive, non-toxic, and easily processed. Different from traditional film TEGs obtained by connecting p-type and n-type strip film through the connection way of "electrical series, thermal parallel" with conductive silver paste and metal electrodes (copper, silver, etc), a new type of single-piece film TEGs with lower contact resistance has been reported, which were prepared by patterned cutting after printing or vapor doping to create p-type and n-type TE legs in films[28,32,33].

Single CNT has remarkable electrical properties. However, these outstanding properties have remained elusive in macroscopic films[17]. Previously, various solvents have been studied to prepare well-arranged CNT fibers[34,35], CNT liquid crystals[17,36], and CNT films (post-treatment)[15,37]. In this work, we reported a material process approach, containing a developed floating catalyst chemical vapor deposition

(FCCVD) with capillary compression assistant alignment and a compressing process, to prepare macroscopic CNT films with highly ordered CNT assemblies (Fig. 1). The newly developed process could produce CNT films at large scales, which could shorten the preparation process of high-quality CNT films as well. The environment-friendly solvent EtOH was used in this work. Besides of the high CNT alignment, the achieved CNT films have a high density and a high purity (low defects and low level of amorphous carbon), resulting in a high electrical conductivity of 2.17 MS m$^{-1}$ and subsequently a high p-type $PF$ of 9.31 mW m$^{-1}$ K$^{-2}$. After being n-type doped with a vapor doping method, the CNT film exhibited a high n-type $PF$ of 6.75 mW m$^{-1}$ K$^{-2}$. The obtained n-type $PF$ is the highest value ever achieved for any n-type CNT materials, which is even higher than that of state-of-the-art n-type inorganic TE materials at room temperature. The reason for the high n-type $PF$ is mainly attributed to the aligned CNTs in the film, which allow the vapor of n-type dopant to permeate deeply inside the film for efficient n-type doping, leading to the suitable position between the $E_f$ and the 1D van Hove singularity for high $S$ of the CNT films. It is also attributed to the high electrical conductivity of 2.94 MS m$^{-1}$ derived from the CNT alignment by the capillary compression as well as the compressing process for high-density CNT films.

As the obtained high-performance CNT films were anisotropic, the computational assistant design was used to fabricate the high-power density curl-able TE generator, which exhibited a high value of 110 W m$^{-2}$ (560.2 μW m$^{-1}$ K$^{-2}$) among flexible (CNT-based) TE devices. A new concept of an intelligent temperature controller (ITC) with a light-emitting diode (LED) alarm was fabricated with integrated cooling and alarming functions together in one device for automated temperature control. The results provide a promising way for high $PF$ and high $k$ CNT films, demonstrating the potential of them in the applications of the automated thermal management of electronic devices, which is different from the use of CNTs in traditional TE cooling applications.

## Results and discussion

We used a developed FCCVD method to synthesize the CNT films. Different from previous literature[8,15], anhydrous ethanol was sprayed at a speed of 3 ml min$^{-1}$ continuously on the CNT aerogel when it was collected on the roller (Fig. 1), named CNT-e films. CNT films without ethanol treatment (CNT-o) were also synthesized for comparison. High-resolution transmission electron microscopy (TEM) images showed that the majority of the CNTs were multi-walled CNTs, as shown in Fig. S1. Then, the CNT-e and CNT-o films were further compressed at a pressure of about 100 MPa to improve the packing density of CNTs for higher electrical conductivity as suggested in the literature[8,33]. Detailed information was described in the supporting information.

Scanning electron microscopy (SEM) and Raman spectroscopy were performed to characterize the CNT alignment in the obtained CNT films. SEM images showed that the obtained CNTs were aligned along the direction that was parallel to the rolling direction. Ethanol-treated CNT films (CNT-e) showed higher alignment than the CNT-o films obviously (Fig. 2a and b). The result was consistent with that obtained from the Raman spectra (Fig. S2) which also showed that both CNT-e and CNT-o had good quality with a high ratio of 21.6 and18.3 for the intensity of graphic structure/the intensity of defect structure ($I_G/I_D$)[33,38]. The TGA curves of CNT-o films and CNT-e films are shown in Fig. S3. The reason for the higher alignment in CNT-e films was attributed to the self-assembly of CNTs induced by the capillary action of ethanol. As shown in the insert image in Fig. 2, the capillary force was derived from the pressure difference ($\Delta P$) between the pressure of air ($P_a$) and the pressure of ethanol ($P_e$) according to the Laplace equation[39]:

$$\Delta P = P_a - P_e = \frac{\gamma}{r} = \frac{2\gamma \cos\theta}{D} \tag{1}$$

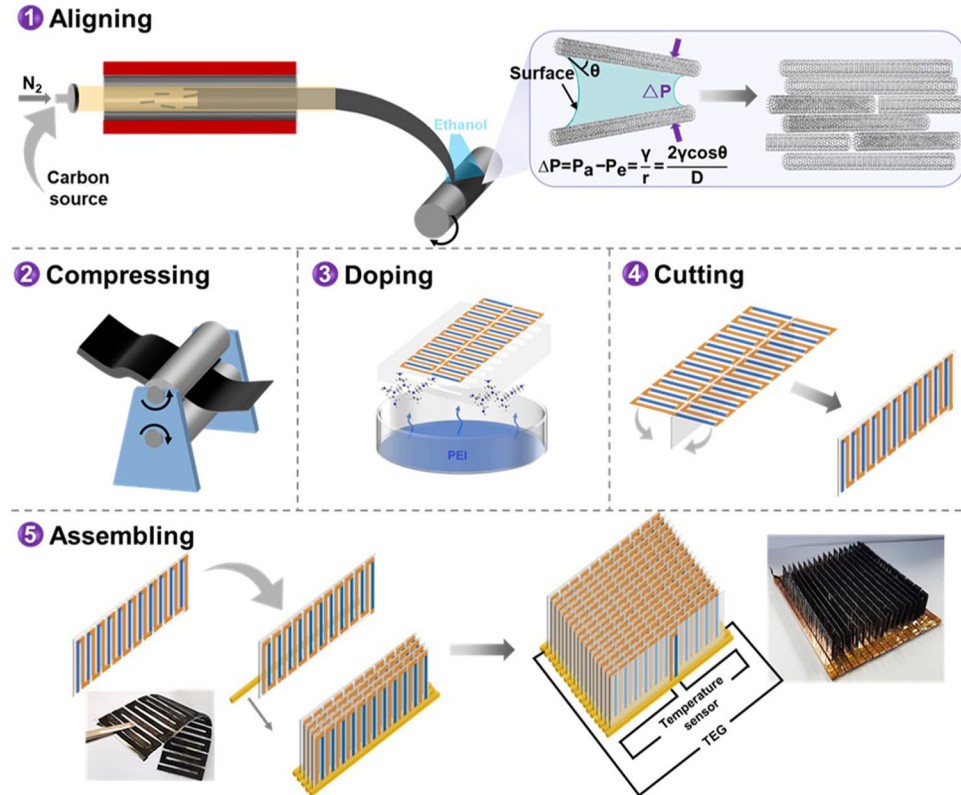

**Fig. 1 | Sample preparation process.** Illustration of the high-performance macroscopic CNT film preparation process and the fabrication process for intelligent temperature controller (ITC).

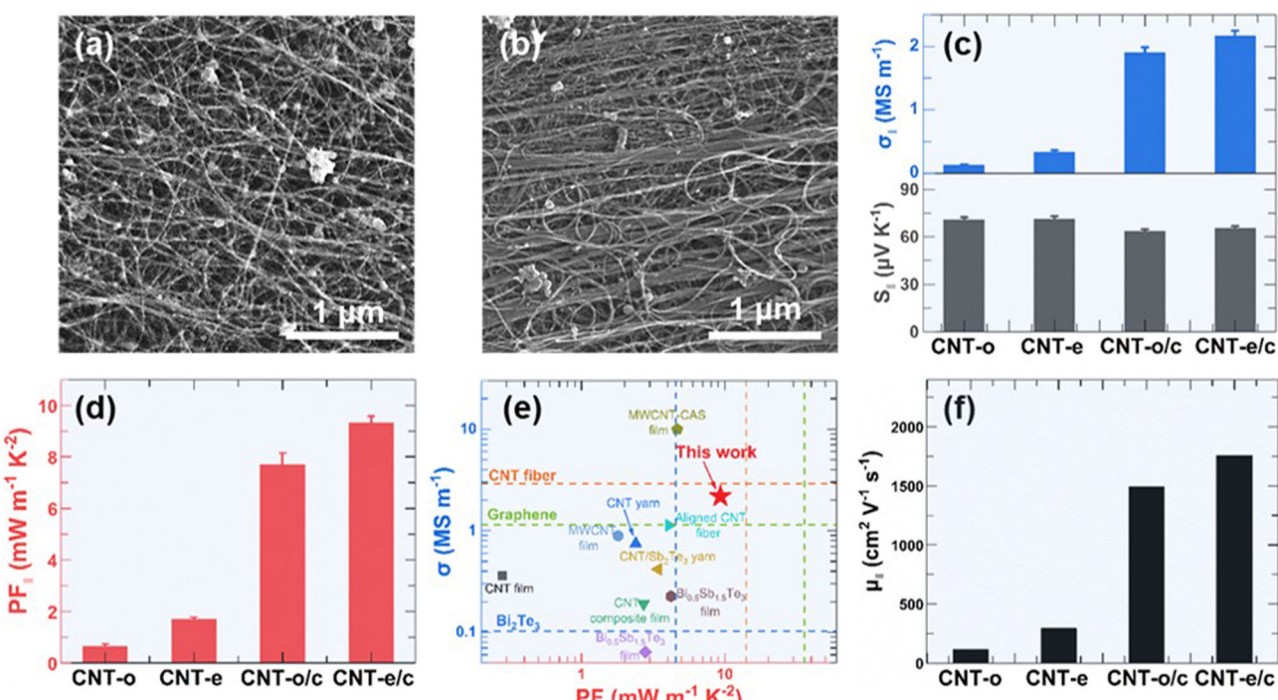

**Fig. 2 | Self-assembly alignment and compression effects on the thermoelectric performance of CNT films.** SEM image of CNT-o/c films (**a**) and CNT-e/c films (**b**). **c** The electrical conductivity ($\sigma_\parallel$), Seebeck coefficient ($S_\parallel$) in the parallel direction of CNT-o films, CNT-e films, CNT-o/c films, and CNT-e/c films. **d** The power factor ($PF_\parallel$) in the parallel direction of CNT-o films, CNT-e films, CNT-o/c films, and CNT-e/c films. **e** Comparison of reported PF values with $\sigma$ for various CNTs and conventional TE materials. **f** The weight mobility in the parallel direction ($\mu_\parallel$) of CNT-o films, CNT-e films, CNT-o/c films, and CNT-e/c films.

where $\gamma$ was the ethanol surface tension and r was the radius of curvature of the ethanol–air interface. D was the distance between two CNTs, $\theta$ was the contact angle of ethanol on the CNT surface. The capillary force and the strong van der Waals force would effectively move the CNTs together to near-ideal graphitic spacing[40,41]. Other alcohol solvents might also show a similar effect on the arrangement of CNTs, various solvents have been studied in literature to prepare well-arranged CNT fibers[34,35], CNT liquid crystals[17,36], and CNT films[15,37].

We further compressed the CNT films at a pressure of about 100 MPa to improve their electrical properties. The compressing process maintained the good quality and good alignment of CNTs as shown in the Raman spectra (Fig. S4). SEM images showed that the compressed CNT-e films (CNT-e/c) had higher alignment than the compressed CNT-o films (CNT-e/c) (Fig. S5). Figure 2c shows that the electrical conductivity in the direction parallel to the rolling direction of CNT-e film ($\sigma_{\parallel(CNT-e)}$) was about 2.5 times higher than the $\sigma_{\parallel(CNT-o)}$ of CNT-o films. The schematic of the anisotropic electrical conductivity measurements for CNT films is shown in Fig. S6. While the Seebeck coefficient in the parallel direction ($S_{\parallel(CNT-e)}$) was almost the same with $S_{\parallel(CNT-o)}$. The positive Seebeck coefficients indicated both the CNT-e and CNT-o were p-type materials. After compressing, the electrical conductivity of CNT-e/c film in the parallel direction ($\sigma_{\parallel(CNT-e/c)}$) significantly increased by 6.6 times, reaching up to 2.17 MS m$^{-1}$. The sheet resistance of CNT-e/c films in the parallel direction measured by the van der Pauw method was consistent with the sheet resistance calculated from the electrical conductivity measured by SBA-458, as shown in Table S1. While the Seebeck coefficient in the parallel direction of CNT-e/c film ($S_{\parallel(CNT-e/c)}$) was nearly the same with $S_{\parallel(CNT-e)}$. Similarly, the electrical conductivity of CNT-o/c film ($\sigma_{\parallel(CNT-o/c)}$) increased with the maintained Seebeck coefficient ($S_{\parallel(CNT-o/c)}$). The detailed electrical properties as a function of compressing time for CNT-e/c and CNT-o/c films are shown in Fig. S7. Figure S8 shows the densities of CNT-e film before and after compressing. The density of CNT-e films was about 0.8 g cm$^{-3}$, which increased to ~2.1 g cm$^{-3}$ for CNT-e/c films. The results indicated that the electrical conductivity was proportional to the density of the CNT films. Similar trend also observed in previous literature[8,33].

The different behavior in electrical properties during the compressing process between CNT-e and CNT-o films was attributed to the anisotropic electrical conductivity and isotropic Seebeck coefficient as suggested in previous literature[8,42]. The increase of electrical conductivity with maintained Seebeck coefficients of CNT films after compressing was due to that the physical compressing process only led to the decrease of the voids in the CNT films with little influence to the chemical environment of CNTs, thus resulting in the decoupling of the electrical conductivity and the Seebeck coefficient[8,33,43]. Ultraviolet Photoelectron Spectroscopy was performed with gold as a reference to identify the work function change of CNT-e and CNT-e/c films (Fig. S9). The obtained work function of CNT-e film and CNT-e/c film exhibited similar values, indicating that the chemical environment of the CNT remained constant (the Fermi energy level is unchanged) before and after compressing. During the compressing process, the key factor affecting the TE performance was the change in the porosity of the CNT films which has been well-discussed in previous works[8,33,43,44]. A detailed analysis has been provided in the supporting information (Fig. S7), where the thickness of the CNT films decreased with the compressing time. For the CNT films with similar porosity, the TE performance would be independent of the thickness as demonstrated by ref. 45. The thicknesses of the CNT film before compressing in this work were in the range of 12.67 ± 0.25 μm and 5.73 ± 0.21 μm for CNT-o films and CNT-e films, which were in the range of 1.12 ± 0.05 μm and 0.91 ± 0.03 μm after compressing (Fig. S10). The decoupling of the two TE parameters then led to a high maximum p-type $PF$ in the parallel direction of CNT-e/c films ($PF_{\parallel(CNT-e/c)}$), up to 9.31 mW m$^{-1}$ K$^{-2}$ (Fig. 2d). This high $PF$ value was comparable to state-of-the-art CNT-

based and conventional TE materials (Fig. 2e)[8,15,33,34,43], which was over two times higher than that of commercially used p-type TE materials Bi$_2$Te$_3$ (~4.5 mW m$^{-1}$ K$^{-2}$)[11].

We calculated the weighted mobility ($\mu$) of the CNT films according to literature[46], which helped us to understand the variation of the electrical properties during the capillary compression-assisted alignment and the compressing process. The detailed calculation was described in the supporting information. Figure 2f showed that the weighted mobility in the parallel direction ($\mu_\parallel$) was in the order of $\mu_{\parallel(CNT-e/c)} > \mu_{\parallel(CNT-o/c)} > \mu_{\parallel(CNT-e)} > \mu_{\parallel(CNT-o)}$. The results indicated that both the alignment and the compressing process benefit the increase of $\mu_\parallel$ in the CNT films, thus leading to the significant improvement of $\sigma_\parallel$ and $PF_\parallel$.

The TE properties of CNT films in the direction perpendicular to the rolling direction have also been characterized (Fig. S11), which were lower than those in the parallel direction. The maximum $\sigma_\perp$ and $PF_\perp$ were only 0.51 MS m$^{-1}$ and 2.21 mW m$^{-1}$ K$^{-2}$, respectively, which were obtained in CNT-e/c films.

We used a vapor treatment method to turn the p-type CNT-e/c and CNT-o/c films into n-type as illustrated in Fig. S12. It was different from the solution doping method by mixing the dopant with CNTs[43,47], which minimized the disturbance by n-type dopants to the aligned CNTs in the films. The detailed doping process was described in the supporting information. A commercially available n-type dopant, poly(ethylene imine) (PEI), was used in this work to check the proposed concept of the n-type doping method. Figure 3a shows the $S_{\parallel(CNT-e/c)}$ and the $\sigma_{\parallel(CNT-e/c)}$ as a function of the PEI vapor treatment time. After being treated by PEI for 10 min, the $S_{\parallel(CNT-e/c-PEI)}$ changed from +64.7 μV K$^{-1}$ to −51.7 μV K$^{-1}$. The negative $S_\parallel$ demonstrated that the PEI-doped CNT-e/c film becomes n-type. When increasing the treatment time, the $S_{\parallel(CNT-e/c-PEI)}$ decreased, which became −44.6 μV K$^{-1}$ after the film being treated for 120 min. While the $\sigma_{\parallel(CNT-e/c-PEI)}$ increased with the treatment time, which reached up to 3.16 MS m$^{-1}$ after 120 min. The decrease of the $S_{\parallel(CNT-e/c-PEI)}$ and the increase of the $\sigma_{\parallel(CNT-e/c-PEI)}$ were attributed to the gradual increase of n-type carrier concentration with PEI treatment time[33,43,47,48].

It was interesting to notice that the $S_{\parallel(CNT-o/c-PEI)}$ (−38.14 μV K$^{-1}$ after being treated by PEI for 120 min) in Fig. 3b was smaller than $S_{\parallel(CNT-e/c-PEI)}$ although the above results in Fig. 2c showed that the p-type $S_{\parallel(CNT-o/c)}$ and $S_{\parallel(CNT-e/c)}$ were nearly consistent with each other. The reason should be attributed to the different $E_f$ positions related to the 1D van Hove singularity, caused by different doping levels. Previous literature demonstrated that shifting the position of the $E_f$ to the 1D van Hove singularity of CNTs by tuning the doping level could not only lead to different Seebeck coefficients but also the simultaneous improvement of the Seebeck coefficient and the electrical conductivity of CNT films[34,47,48]. The different doping levels in CNT-e/c-PEI films and CNT-o/c-PEI films should be due to the different order parameters of CNTs, which allowed different amounts of PEI vapor to permeate inside the films (Fig. 3c and d). In the CNT-o/c films, the CNT bundles were packed less ordered, which would prevent PEI vapor from permeating deeply inside the films, thus leading to a low n-type Seebeck coefficient of $S_{\parallel(CNT-o/c-PEI)} = -38.14$ μV K$^{-1}$ for the inefficient n-type doping. While the CNT bundles in CNT-e/c films were packed more ordered, which was in favor of the deep permeation of PEI vapor inside the films for efficient n-type doping with a high n-type Seebeck coefficient of $S_{\parallel(CNT-e/c-PEI)} = -44.6$ μV K$^{-1}$. The order parameter induced different doping levels has been illustrated in Fig. 3c and d. Similar to this work, using aligned CNT films as electrodes was demonstrated to be an effective strategy to improve the performance of electrochemical reactions since they were in favor of the redox ions permeating through the thick CNT electrodes[49,50].

The $\sigma_{\parallel(CNT-o/c-PEI)}$ was 2.85 MS m$^{-1}$ after being treated by PEI for 120 min (Fig. 3b), which was also smaller than $\sigma_{\parallel(CNT-e/c-PEI)}$. It was mainly attributed to the lower order parameter that leads to lower

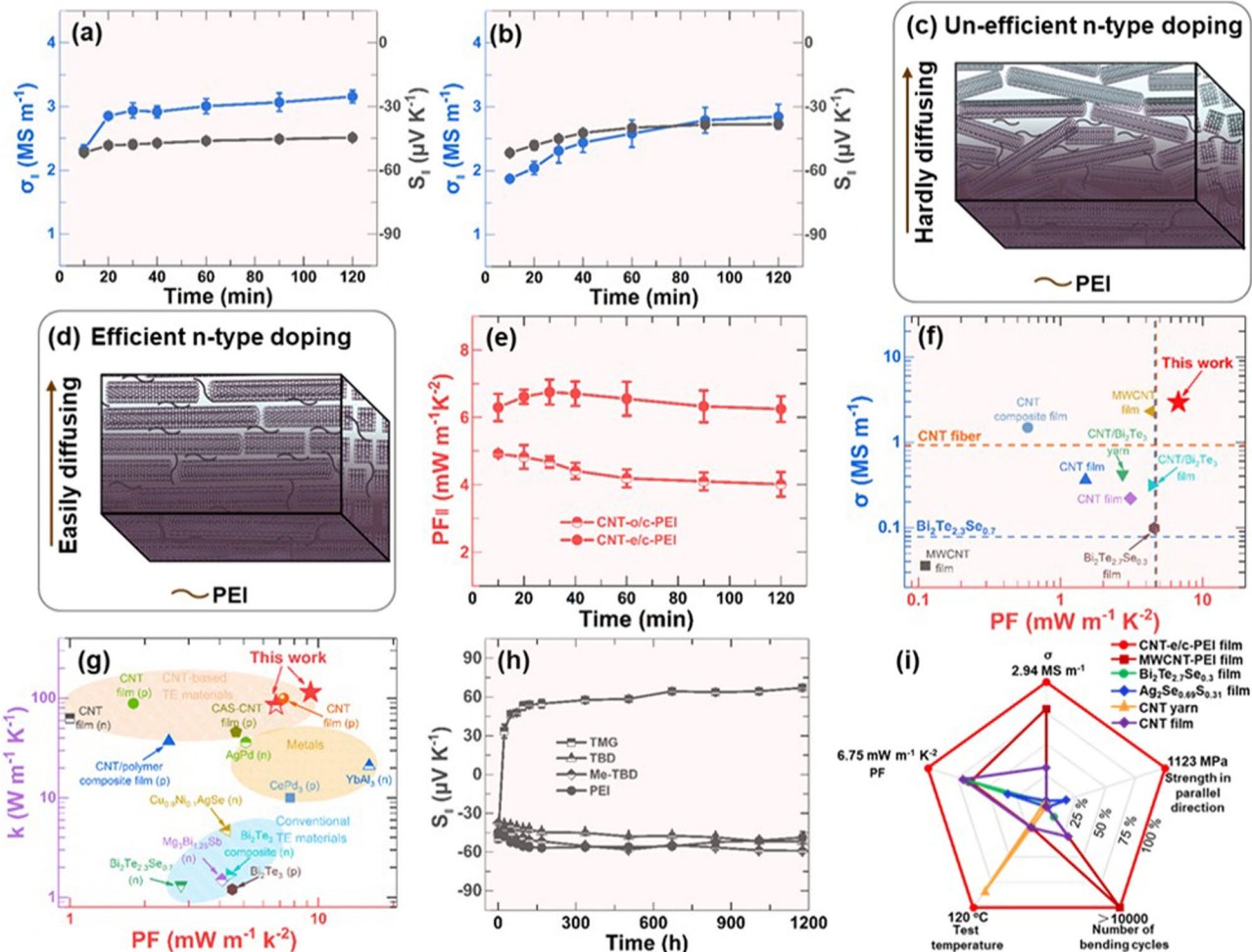

**Fig. 3 | The thermoelectric performance of n-type CNT films prepared by vapor doping.** The electrical conductivity ($\sigma_\parallel$), Seebeck coefficient ($S_\parallel$) in the parallel direction of CNT-e/c films (**a**) and CNT-o/c films (**b**) as a function of the PEI vapor treatment time. Illustration of different doping levels of CNT-o/c-PEI films (**c**) and CNT-e/c-PEI films (**d**). **e** The power factor ($PE_\parallel$) in the parallel direction of CNT-e/c films and CNT-o/c films. **f** Comparison of reported $PF$ values with $\sigma$ for various CNTs and conventional TE materials. **g** Comparison of reported $PF$ values with the thermal conductivity ($\kappa$) for various CNT films, metals, and conventional TE films. **h** Long-term stability of n-type dopants doped CNT-e/c films at 120 °C in air. **i** Comparison of comprehensive performance of CNT-e/c-PEI films including $\sigma_\parallel$, $PE_\parallel$ flexibility, mechanical strength and temperature stability.

charge carrier mobility in the CNT-o/c-PEI films. The maximum weighted mobility of CNT-e/c-PEI films was 1703 cm$^2$ V$^{-1}$ s$^{-1}$, which was about 1.3 times higher than that of CNT-o/c-PEI films (Fig. S13 and Table S2). The maximum n-type $PF_\parallel$ of CNT-e/c-PEI films was achieved to be 6.75 mW m$^{-1}$ K$^{-2}$, which was about 1.37 times higher than that of CNT-o/c-PEI films (Fig. 3e). The electrical conductivity, the Seebeck coefficient and the $PF$ of the front and back sides of the CNT-e/c-PEI films were almost consistent as shown in Fig. S14. The results indicated that the n-type CNT films were uniformed doped by the PEI vapor. The content of PEI in the CNT film was roughly calculated to be about 10.3% according to the weight variation of CNT-e/c-PEI films between 158-350 °C, as shown in Fig. S15[51,52]. The TE properties of PEI-doped CNT films in the perpendicular direction have also been characterized (Fig. S16). The maximum $\sigma_\perp$ values were in the order of $\sigma_{\perp(CNT-o/c-PEI)} > \sigma_{\perp(CNT-e/c-PEI)}$. The maximum $PF_\perp$ values were in the order of $PF_{\perp(CNT-o/c-PEI)} > PF_{\perp(CNT-e/c-PEI)}$.

We compared the maximum n-type $PF_{\parallel(CNT-o/c-PEI)}$ with previous literature (Fig. 3f). The maximum n-type $PF_\parallel$ of 6.75 mW m$^{-1}$ K$^{-2}$ for PEI doped CNT-e/c films was the highest value ever achieved for any CNT materials, which was even ~1.5 times higher than that of recently reported state-of-the-art n-type inorganic Bi$_2$Te$_3$ films (~4.6 mW m$^{-1}$ K$^{-2}$)[31]. Note that the high $PF$ (9.31 mW m$^{-1}$ K$^{-2}$ for p-type and 6.75 mW m$^{-1}$ K$^{-2}$ for n-type) in this work could be achieved in macroscopic samples (36 cm × 12 cm), which was different from the 2D materials that exhibit the high $PF$ typically in µm-scale[21,22]. The scalability of these high-performance CNT films would ease the fabrication of TE devices for the practical applications.

In addition, the alignment and the compressing process also resulted in the enhancement of the thermal conductivity of CNT films in the parallel direction ($k_\parallel$) because the alignment increased the mean free path of phonons[43] and the compressing process reduced the thermal insulating voids[8,33]. The $k_{\parallel(CNT-e/c)}$ and $k_{\parallel(CNT-e/c-PEI)}$ for the films with the maximum $PF_{\parallel(CNT-e/c)}$ and $PF_{\parallel(CNT-e/c-PEI)}$ were 114.3 W m$^{-1}$ K$^{-1}$ and 86.23 W m$^{-1}$ K$^{-1}$, respectively (Fig. S17 and Table S3). The attempts to detect the out-of-plane thermal conductivity of CNT films failed due to the ultra-thin thickness (~1 µm) of the prepared CNT films, making their out-of-plane thermal conductivity difficult to measure. Similar results have been reported in the previous literature[29,47,48]. The in-plane thermal conductivity in the direction perpendicular to the CNT orientation ($k_\perp$) would be similar to the out-of-plane thermal conductivity ($k_{out}$), due to the good alignment of the CNTs, as shown in Fig. S18. The $k_\perp$ values of CNT films are shown in Fig. S17b. Both the high $k$ and the high $PF$ were required in active cooling, which were not easy to be obtained simultaneously in previous materials. As shown in Fig. 3g, conventional CNT films often had high $k$ but low $PF$, while conventional inorganic materials often had

high $PF$ but low $k$ at 300 K. The $k_{eff}$ values of the p-type CNT-e/c films and the n-type PEI doped CNT-e/c films were calculated at the given conditions of $\Delta T = 1$ K at 300 K, which were 532.78 W m$^{-1}$ K$^{-1}$ and 388.98 W m$^{-1}$ K$^{-1}$, respectively. The $\kappa_{eff}$ of our CNT films exceeded those of CePd$_3$ and conventional TE materials (Table S4).

In the meanwhile, the $ZT$ value increased by 1.2–2.5 times for the CNT films after compressing and vapor doping as shown in Fig. S19. The reason for the increase of $ZT$ was attributed to the increase of the electrical conductivity/thermal conductivity ratio since the compressed CNT films had different scattering effects on electrons and phonons due to their different mean free paths. Similar results have also been obtained in previous works[8,33,43]. The maximum $ZT$ of the CNT-e/c films was comparable to the state-of-the-art reported CNT films as shown in Table S5[8,15,29,33,53–55]. Although the $ZT$ value of CNT films was not high, the excellent electrical and mechanical properties of CNTs make them promising candidates for flexible TE materials. Research on improving the TE properties of CNT-based materials was still very attractive recently[8,15,24,29,33,38,43–45,47,48,56]. In the meanwhile, previous literature suggests that $ZT$ should not be the only parameter to evaluate a TE material. High power factor CNT films were still highly desired to fabricate TEGs with high output power density for the practical applications when the waste heat was abundant and released freely[2,8,24,33,38,43,44].

We tested the stability, flexibility, and the mechanical strength of the n-type CNT-e/c-PEI films for practical applications. Previous work demonstrated that PEI-doped CNT films had great stability at room temperature, which maintained their $S$ at nearly 100% for even 120 days[33]. The temperature stability of CNT-e/c-PEI films at 120 °C was tested here. Figure 3h showed that the $S_{\parallel (CNT-e/c-PEI)}$ increased at the very beginning of heating, which was attributed to the increase of efficient n-type doping at high temperatures as suggested in previous literature[56]. The $S_{\parallel (CNT-e/c-PEI)}$ maintained a high value comparable to/ even a little bit larger than its original value after CNT-e/c-PEI films were heated at 120 °C for 1200 h. For comparison, literature reported highly stable n-type dopants[56], 1,5,7-trizazabicyclo[4.4.0]dec-5-ene (TBD), 7-methyl-1,5,7-trizazabicyclo[4.4.0] dec-5-ene (Me-TBD) and 1,1,3,3-tetramethylguanidine (TMG), were used to dope CNT-e/c films using the same vapor doping method. The $S_{\parallel}$ and $\sigma_{\parallel}$ of these dopants doped CNT-e/c films as a function of vapor treatment time were shown in Fig. S20. The $S_{\parallel}$ of TMG doped CNT-e/c films dropped fast at 120 °C, which turned into p-type after only 24 h. While the $S_{\parallel}$ values of TBD and Me-TBD doped CNT-e/c films showed great stability similar with $S_{\parallel (CNT-e/c-PEI)}$ under the same vapor doping conditions.

To fully evaluate the performance of the prepared CNT films, we compared the $\sigma_{\parallel}$, $PF_{\parallel}$, high temperature stability, flexibility, and mechanical strength of both CNT-e/c and CNT-e/c-PEI films with previously reported TE films[15,31,33,56]. Figures 3i and S21 indicated that both CNT-e/c and CNT-e/c-PEI films were high-performance multiple functional films, which had not only high basic TE properties, such as $\sigma_{\parallel}$ and $PF_{\parallel}$, but also excellent flexibility, great mechanical strength, and good temperature stability for practical applications. The strain-stress results were shown in Fig. S22, which indicated that both CNT-e/c and CNT-e/c-PEI films had high strength values of 1240 MPa and 1123 MPa, respectively, in the parallel direction. The strength values of CNT-e/c and PEI-doped CNT-e/c films in the perpendicular direction were much lower than that in the parallel direction, which were only 144 MPa and 137 MPa, respectively. Figure S23 showed that the electrical properties of CNT-e/c and CNT-e/c-PEI films in the parallel direction were nearly constant after being bent 10,000 times. These results showed that CNT-e/c and CNT-e/c-PEI films were promising TE materials with both high TE properties and high mechanical properties required for potential practical applications.

After having the high-performance flexible TE films, a TE generator was fabricated to demonstrate the heat-to-electricity conversion ability of these films. The shape-property relationship was studied

in advance to improve the output power density of TEGs. Single-piece structure TE device was designed since it would minimize the consumption of the generated power of the internal resistance by reducing electrical contact resistance at the p-n junction[28,32,33].

Computational design was performed to assist in the fabrication of the single-piece TE device. Firstly, the polarity of the connection bridge should be the same with the TE module (matching) which had higher thermal conductivity when assuming the geometry of the p-n modules was the same. As shown in Fig. 4a, the connection bridge might also generate electricity due to the temperature difference between the cold side of the p- ($T_{cold-p}$) and n-type ($T_{cold-n}$) modules since the thermal conductivities of the p- and n-type materials were different. The generated voltage would benefit the TE generator when the voltage direction is agreed with that of the TE generator. Otherwise, it would reduce the performance of the TE generator. The theoretical calculation was performed with COMSOL finite element simulation to show the influence of different polarity bridges on the output voltage of the TE generator. Figure 4b shows that the different thermal conductivity of the p- and n-type materials would lead to different temperatures at the cold side of the TE modules, thus generating TE voltage at the connection bridge. The thermal conductivity difference was $k_{\parallel (CNT-e/c)}$ : $k_{\parallel (CNT-e/c-PEI)} = 1.34$ for CNT-e/c and CNT-e/c-PEI films, which would result in a TE voltage difference of 0.46 mV between the TE generator with polarity matched connection bridge and the TE generator with polarity mismatched connection bridge when the TEGs with one pair of p-n modules were attached on a heat source with a temperature of 100 °C. The TE voltage difference was not negligible, especially when increasing the number of p-n modules. In the meanwhile, the larger the thermal conductivity difference was, the higher the temperature difference between the two sides of the connection bridge would be, subsequently leading to a larger TE voltage difference of the device.

Secondly, the width and the length needed to be optimized for high output power density TEGs. As the TE properties of both CNT-e/c and PEI-doped CNT-e/c films were anisotropic, three types of TEGs have been proposed: parallel structure (the length of TE modules parallel to the rolling direction), perpendicular structure (the length of the TE modules perpendicular to the rolling direction) and annular structure, as shown in Fig. 4c. Assuming the temperature of the hot source was 100 °C and the environmental temperature was 25 °C, the length and the width of the three types of TE generator were optimized with the filler factor of 0.5 (Fig. S24). Fill factor ($FF$) was defined as: $FF = D/(D + W)$, where $D$ was the distance between p- and n-type modules and $W$ was the width of the TE modules. The maximum output powers, in theory, were in the order of parallel structure (33.7 μW) > annular structure (20.3 μW) > perpendicular structure (18.8 μW) for a constant D + W of 9 cm (Fig. 4d). The optimized length $x$ width for the parallel structure was 25 mm × 2 mm, which was 15 mm × 2 mm for the perpendicular structure and annular structure devices. With the optimized width and length, the $FF$ was then discussed theoretically. Figure S25 shows that the output power increases with the increase of $FF$.

TE generator with the parallel structure (par-TEG) was fabricated. It contained 11 pairs of p-n modules with the length of 25 mm and the width of 2 mm at an $FF$ of 0.5 (Fig. 4e), due to the limitation of lab conditions. The fabricated TE generator had good flexibility with no need for extra metal electrodes which could be rolled easily into a cylinder (Fig. 4e)[32,33]. The par-TEG exhibited an open-circuit voltage of 81.5 mV and a short-circuit current of 1.4 mA at a temperature difference of 85 K (Fig. 4f). The maximum power output of the par-TEG was 29.7 μW when the loading resistance equaled to the inner resistance of 56 Ω. The theoretical open-circuit voltage and the theoretical output power were calculated as shown in Fig. S26. The open-circuit voltage and maximum power output of the SP-TEG are slightly lower than theoretical values. The power density by weight[57] and the power

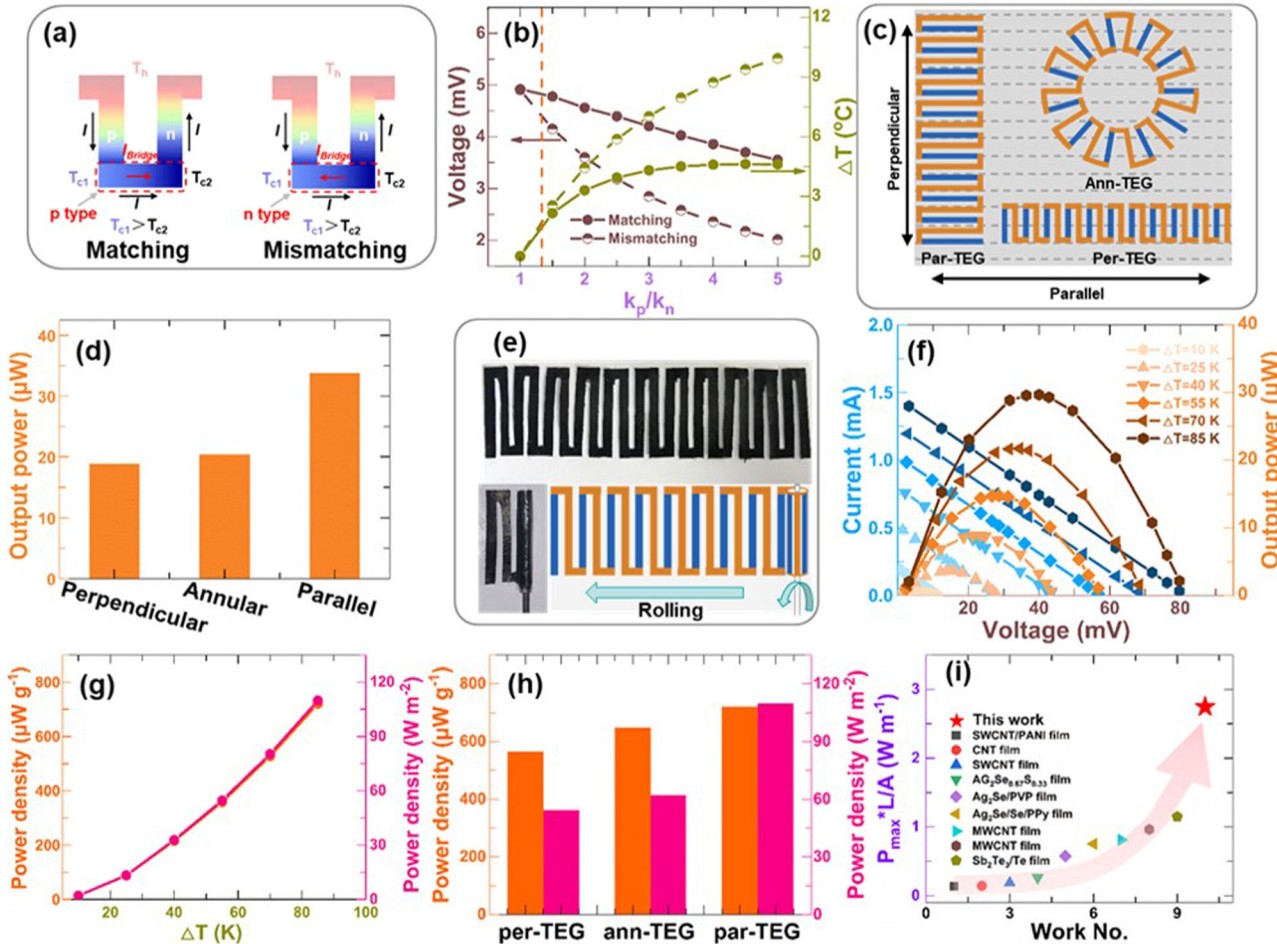

**Fig. 4 | Output performance of single-piece thermoelectric generators prepared by CNT films. a** Schematic diagram of thermal conductivity difference of thermoelectric modules resulting in electricity generation of the connecting bridge. **b** Theoretical calculation results of output voltage and temperature difference varying with $k_p/k_n$ of one pair of TE modules. **c** Illustration of par-TEG, per-TEG, and ann-TEG in aligned CNT film. Theoretical calculation of the maximum output power (**d**) of par-TEG, per-TEG, and ann-TEG. **e** Image of the rolling process for par-TEG. **f** The voltage-current curves of par-TEG at different temperature differences. **g** The areal and weight output power of par-TEG at different temperature differences. **h** The maximum areal and weight output power of p par-TEG, per-TEG, and ann-TEG at 85 K. **i** Comparison of the maximum areal output power of TE generator with flexible TE generator in literature.

density by area were calculated with the maximum of 720 µW g⁻¹ and 110 W m⁻² at the temperature difference of 85 K (Fig. 4g). For comparison, the TEGs with the perpendicular structure (per-TEG) and annular structure (ann-TEG) were all fabricated (Fig. 4f). The TE performances of per-TEG and ann-TEG were lower than that of par-TEG as shown in Fig. S27. The maximum power densities for per-TEG and ann-TEG were 54 W m⁻² and 62 W m⁻², respectively (Fig. 4h).

We compared the power output density of par-TEG with that of the flexible TEGs in the literature. The maximum power output density was normalized according to the literature[30,57–59]. Figure 4i showed that the normalized power output density was as high as $P_{max}*L/A = 2.75$ W m⁻¹ at a temperature difference of 85 K, which was above the previously reported values for both organ TEGs. For example, He and Shi et al. reported that the normalized power output density for a flexible PVDF/Ta₄SiTe₄ composite film-based TE generator was <0.15 W m⁻¹ [58]. Lin et al. reported that the normalized power output density for flexible CNTs-Te-PEDOT: PSS composite film-based TE generator was <0.27 W m⁻¹ [57]. Shi and Chen et al. reported normalized power output density values of <0.1 W m⁻¹ and <0.3 W m⁻¹ for full-inorganic devices Ag₂S₀.₅Se₀.₅/Pt-Rh and Ag₂Se₀.₆₇S₀.₃₃/Pt-Rh, respectively[30,59]. The results demonstrated the good heat-to-electricity conversion ability of the par-TEG fabricated with the developed high *PF* CNT films.

The output power stability of the TE generator is shown in Fig. S28. The output power of par-TEG showed good stability in the air at room temperature, which retained nearly 95% of its output power in 30 days. The output power of the par-TEG, which was stored in water, decreased with time. After 30 days, it maintained only 20% of the original value. While the par-TEG was encapsulated with polyethylene terephthalate (PET) films, it could maintain above 90% of the initial value after being kept under water for over 30 days. Although the prepared SP-TEG/par was not able to maintain 80% of the original output power after 30 days while being immersed in water directly as reported in the previous literature[60], the encapsulated SP-TEG/par exhibited superior stability of the output power in water.

We assembled the par-TEGs into an ITC with an LED alarm as a concept for potential applications in the thermal management of electronic devices, which integrated the cooling and alarming functions in one device. Figure 5a shows that one double-sided par-TEG contains 22 pairs of p-n modules with a single-piece structure, which then attached on a PET thin film substrate. The ITC was fabricated by assembling each double-sided par-TEG between two copper strips. Weighting paper with suitable shape was used to make sure that it was electrically insulating between the copper strips and the double-sided par-TEGs. The ITC contained a total of 16 double-sided par-TEGs, which

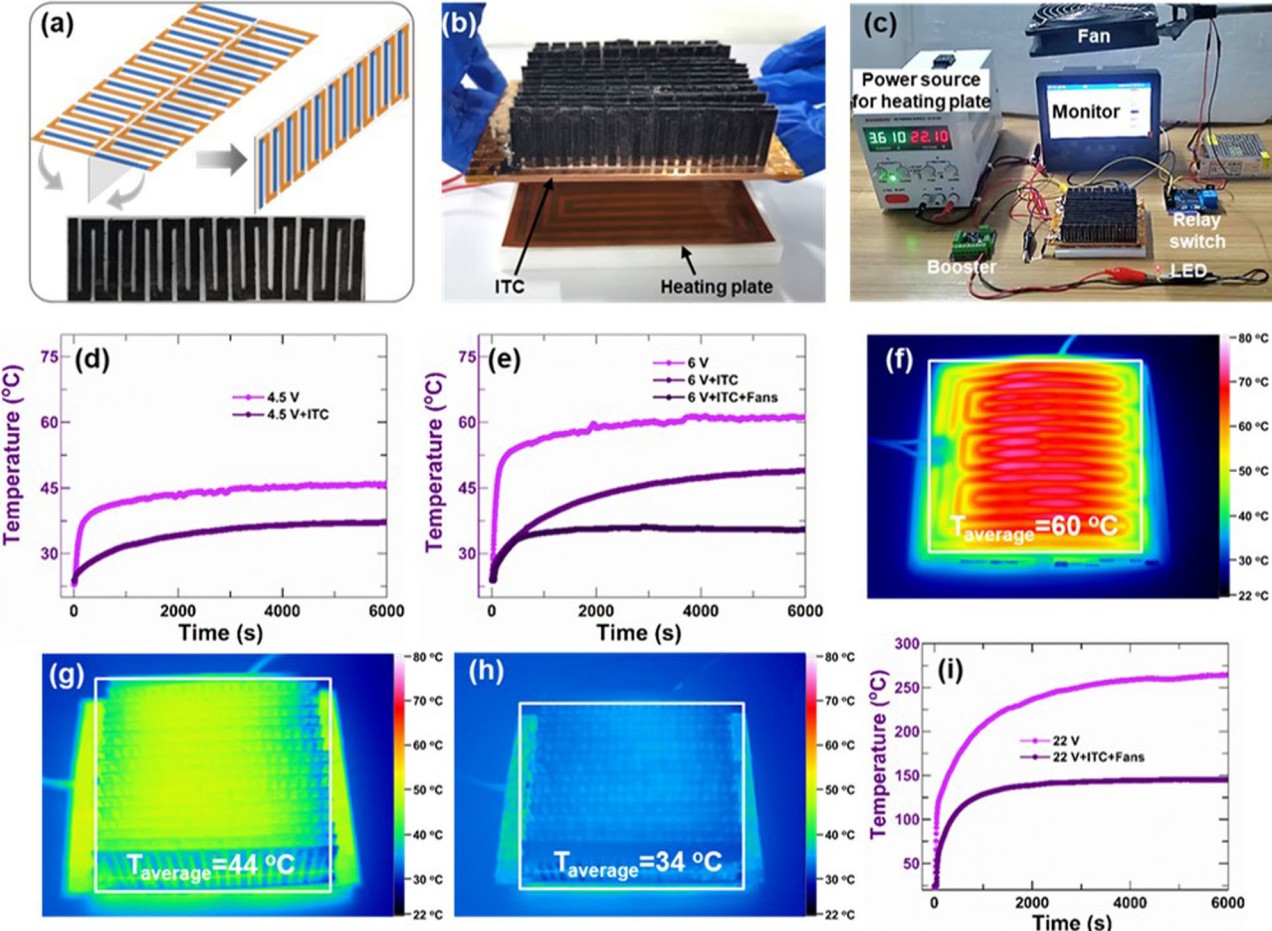

**Fig. 5 | Application of intelligent temperature controller (ITC). a** Schematic diagram double-sided par-TEG. **b** The image of ITC placed on an electrically heating plate. **c** The alarm image of ITC under over-high temperature. **d** The temperature changes of the heating plate under a voltage of 4.5 V without and with ITC. **e** The temperature changes of the heating plate under a voltage of 6 V without and with ITC. The infrared images of the heating plate without ITC (**f**), the heating plate with ITC (**g**), and the heat plate with ITC plus the forced heat dissipation by a fan (**h**). **i** The temperature changes of the heating plate under a voltage of 22 V without and with ITC.

were then placed on an electrically heating plate to test the temperature-controlling ability of the ITC (Fig. 5b).

The whole testing system was shown in Fig. 5c, which contained a fan, a power source and a temperature monitor, etc. The ITC could work in three modes, namely low temperature, high temperature, and over-high temperature mode. At low temperature mode, the heat could dissipate through the ITC for cooling. For example, when the plate was heating at a voltage of 4.5 V, it had a temperature of ~46 °C. After placing the ITC on the heating plate, the temperature decreased and stabilized at ~37 °C since the heat was dissipated through the ITC due to the good thermal conductivity of the CNT modules (Fig. 5d). At high temperatures when the plate was heating at a voltage of 6 V, the heating plate was ~61 °C without ITC. After placing the ITC on the heating plate, although the temperature decreased to ~48 °C, it was still higher than the setting temperature of 37 °C (Fig. 5e). Then, the ITC provided an increased output voltage as a signal to turn on the fan for forced heat dissipation at the wind flow rate of 2 m s⁻¹. Therefore, the temperature decreased to ~35 °C. Figure 5f, g, and h show the infrared images of the heating plate without ITC, the heating plate with ITC, and the heat plate with ITC plus the forced heat dissipation by a fan. For over-high temperatures, when a voltage of 22 V was supplied to the heating plate, the temperature would increase to ~260 °C. The temperature was as high as ~145 °C with the ITC, which was higher than the alarm temperature. Then, the ITC turned the LED on. The results demonstrated the potential applications of high-performance CNT

films in intelligent temperature controlling and alarming for the thermal management of electronic devices.

The CNT film-based TEG was used as a heat sink for passive cooling at a low temperature to save energy, which was different from traditional applications of TEGs in active TE cooling that required power supply all the time. It was also used as a power generator to support an LED temperature alarm at a high temperature to avoid overheating. The high power factor of the CNT films was important for the TEG to generate sufficient output power over the limited contact area of the heat source to support the LEDs for alarming because the output power density of the TEG was proportional to the power factor of CNT films[7,33,43]. A new concept of an ITC with an LED alarm was intentionally proposed, instead of the active TE cooling device, which would extend the application range of the CNT films with a high power factor and a high thermal conductivity in the thermal management of electronic devices.

Macroscopic CNT films with an outstanding n-type power factor of 6.75 mW m⁻¹ K⁻² were synthesized with a material process approach, containing a developed floating catalyst chemical vapor deposition method with capillary compression assistant alignment and a compressing process. The obtained n-type power factor is the highest value ever achieved for any n-type CNT materials, which is even higher than that of state-of-the-art n-type inorganic TE materials at room temperature[31]. The reason for the high n-type power factor was attributed to the high electrical conductivity of 2.94 MS m⁻¹ derived

from CNT alignment by the capillary compression as well as the compressing process. It was also attributed to the high Seebeck coefficient due to aligned CNTs in the film, allowing the vapor of the n-type dopant to permeate deeply inside the film for efficient n-type doping. Besides of the high TE properties, the macroscopic CNT films also had excellent flexibility, great mechanical strength, and good high-temperature stability (120 °C) for practical applications.

The computational assistant design was used to fabricate the high-power density curl-able TE generator since the high-performance CNT films were anisotropic. It exhibited a high normalized power output density of 2.75 W m$^{-1}$ at a temperature difference of 85 K, which was higher than that of previously reported flexible all-inorganic TEGs (<0.3 W m$^{-1}$)[30,57–59]. An ITC was fabricated by assembling these TEGs, which exhibited automated temperature-controlling ability. The outstanding power factor and excellent mechanical properties make these macroscopic CNT films strong candidate for the automated thermal management of electronic devices, which requires a large TE power factor and a large thermal conductivity simultaneously.

## Methods

### Materials
Ethanol was purchased from Tianjin Fuyu Fine Chemical Co., Ltd., China. Thiophene and ferrocene were obtained from Meryer Chemical Technology Co., Ltd., China. Methanol and n-hexane were purchased from Tianjin Zhiyuan Chemical Co., Ltd., China. Polyethyleneimine (PEI) (99 %) was purchased from Macklin Co., Ltd., China. 1,5,7-triazabicyclo[4.4.0]dec-5-ene (TBD) was purchased from Tianjin Heowns Biochemical Technology Co., Ltd., China. 7-methyl-1,5,7-triazabicyclo[4.4.0] dec-5-ene (Me-TBD) was purchased from Bide Pharmatech Co., Ltd., China 1,1,3,3-tetramethylguanidine (TMG) was purchased from Meryer Chemical Technology Co., Ltd., China. All the chemicals were used without further purification.

### Preparation of CNT film
MWCNT was synthesized in a commercial tube furnace (GSL-1700X) at 1500 °C. The precursor solution was made by dissolving the ferrocene catalyst in a mixture of n-hexane, methanol, and thiophene. The solution was injected into the reactor at a rate of 1–2 ml min$^{-1}$ and then carried into the high-temperature zone by N$_2$ at a flow rate of 0.5–1 L min$^{-1}$. The reaction occurred at normal atmospheric pressure. The produced CNT aerogel was driven out by nitrogen and collected at the tube outlet by a roller with a rotational speed of about 6 mm s$^{-1}$. The absolute ethanol was uniformly sprayed at 3 ml min$^{-1}$ during CNT collection to orient the CNTs by self-assembly (CNT-e).

### MWCNT film densifying
A commercial rolling pressing machine (Kejing MSK-2150 China) was used to densify MWCNT films. Two rollers rotated synchronously but in opposite directions. A constant pressure force of -100 MPa was applied during the compression process.

### Doping method
The vapor doping was performed by attaching the CNT films (10 mm × 30 mm × 0.9 μm) on the upper side of a glass-made cylindrical sublimation chamber (40 mm diameter, 70 mm height, Yangzhou Sunflower Glass Instrument Factory). At the same time, a certain amount (0.2 g) of n-type dopants (PEI, Me-TBD, TBD, or TMG) was placed on the bottom of the cylindrical sublimation chamber. The chamber was heated on a hot plate with a sand bath and without a magnetic stirrer at 120 °C under ambient pressure. Therefore, dopant molecules are deposited on the CNT film surface after a certain time of heating (from 10 min to 120 min, Fig. 3, Figs. S16 and S20). After finishing doping, the samples were left to cool down to room temperature naturally in the air. The scheme of the vapor doping process is shown in Fig. S29.

### The finite element analysis for calculating the performances of TEG
The heat transfer in solids and electric currents modules in COMSOL Multiphysics was used to calculate the TE performance of TEGs to optimize the structural design of TEGs. The performances of TEG with the different wide and length of the TE legs as well as the fill factor were simulated by COMSOL Multiphysics simulations. The materials properties used in finite-element analyses are shown in Table S6.

### Preparation of TEG and ITC
TEG of parallel structure (par-TEG), perpendicular structure (per-TEG), and annular structure (ann-TEG) with 11 pairs of p-n legs were fabricated by hand. The length of par-TEG and per-TEG is 90 mm. The length and width of the TE leg for par-TEG are 25 mm and 2 mm, respectively, which are 15 mm and 2 mm, respectively, for per-TEG. The inner diameter and outer diameter are 14 mm and 29 mm for ann-TEG. The length and width of the TE leg for ann-TEG are 15 mm and 2 mm, respectively. The film of CNT was sealed by polyethylene terephthalate (PET) films with a patterned structure, and then the CNT film was treated with PEI vapor. After 30 min doping with 0.2 g PEI at 120 °C, the exposed CNT film was converted from p-type into n-type. After cutting, par-TEG, per-TEG, and ann-TEG were obtained.

An ITC was assembled with 16 pieces of double-sided par-TEG and a temperature sensor of a pair of p-n TE legs. The double-sided par-TEG was attached on a PET thin film substrate (0.2 × 25 × 90 mm). The ITC was fabricated by assembling each double-sided par-TEG between two copper strips (3 × 5 × 12 mm) with adhesive tape. Weighting paper with suitable shape was used to make sure that it was electrically insulating between the copper strips and the double-sided par-TEGs. The par-TEG is connected in series by silver paste.

### Characterization
The electrical conductivity and Seebeck coefficient of samples were measured under Ar protection by using commercial equipment (NETZSCH, SBA-458, Germany). A four-point probe method is designed for the electrical conductivity measurement in the machine. A highly sensitive temperature probe and temperature control system are used with a temperature stability of ± 0.02 K as claimed by the equipment supplier. The measurement errors for the electrical conductivity and the Seebeck coefficient are well-controlled to be 5% and 3%, respectively. The thermal conductivity was measured by using commercial equipment (NETZSCH, LFA 467, Germany) as well. The measurement error of the thermal conductivity is less than 2% as claimed by the equipment supplier.

For each data point, three or more samples were tested to get the real properties of the samples. The error (E) was calculated according to the following equation:

$$E = \sqrt{\sum_{i=1}^{n}(X_i - X_m)^2/n} \text{ and } X_m = \sum_{i=1}^{n} X_i/n$$

where $X_i$ and $n$ are experimental data and the number of samples, respectively.

Scanning electron microscope images were obtained by using FEI Sirion 200. Raman spectra were recorded by a Laser Raman Spectrometer (Thermo Fisher, USA) with an excitation wavelength of 532 nm. The output voltage and output power of devices were measured with a Keithley 2400 Multimeter (Keithley Instruments Inc., USA).

## Data availability
The authors declare that the data supporting the findings of this study are available within the paper and its supplementary information files. Source data are provided with this paper.

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

## Acknowledgements

H.W. acknowledges financial support from the National Natural Science Foundation of China (Grant No. 52276014 and 51888103), the Fundamental Research Funds for the Central Universities, and The World-Class Universities (Disciplines). This work was also supported by the HPC platform and the Instrument Analysis Center, Xi'an Jiaotong University.

## Author contributions

H.W. and K.L. contributed to the manuscript equally. H.W. and K.L. conceived and planned the experiments; K.L. carried out the experiments; H.X., J.P., T.Z., X.D. and J.W. contributed to sample preparation; H.W. performed the analysis and wrote the manuscript; B.C., D.C., contributed to the interpretation of the results; All authors discussed the results and commented on the manuscript.

## Competing interests

The authors declare no competing interests.
