## [Peer Review File · Nature Communications]

Capillary compression induced outstanding n-type thermoelectric power factor in CNT films towards intelligent temperature controllerREVIEWER COMMENTS

Reviewer #1 (Remarks to the Author):

This paper consistently examines the manufacturing method of CNTs for thermoelectric applications, the anisotropic physical properties of p- and n-materials, and the evaluation of curl-type devices. I feel that this is an important insight that will open up previously unexplored possibilities of nanotube structures. Many experiments were performed, detailed manuscripts were written, and several interesting phenomena were discovered. Since the majority of fossil fuels are converted into unused heat, designing highly efficient power generation materials for energy harvesting is an eternal challenge and one of the important issues facing each country. Considering the combination of AI and IoT and the future development of IT technology in recent years, the approach discovered by the authors is very timely, and the content of the research results is also important. This paper is of great interest to readers, as this result is expected to contribute not only to the fields of engineering and materials, but also to the development of business in society and people's life work in the future. I generally agree with the arguments in this manuscript and consider its content to be of a high standard and appropriate for the journal. My small doubts and questions are summarized below. Although this manuscript is already interesting and informative, I will recommend it to this journal once these concerns and issues are addressed.

1 . Change in film thickness due to pressing

I expect to find out more about the extent to which the film thickness changed before and after Scheme 1 compressing. Notably, the prepared film had a thickness > 100 nm, and its TE performance was observed to be independent of the thickness of the CNT film (Energy Environ. Sci., 2017, 10, 2168.). I think this information is necessary for the reader to perform reproducible experiments. In Figure S6, it seems that the thickness of the sample was measured at only one location by SEM. It is more accurate to measure the film thickness of the sample at 200 locations using a film thickness meter instead of using an SEM and check the average value. I think it would be better to include at least the measured data of the main samples in this paper (CNT-o/c films, CNT-e/c films, CNT-o/c-PEI films, CNT-e/c-PEI films) in the ESI.

2 . EtOH treatment

I believe the author is making the claim that the EtOH treatment in Scheme 1 results in a one-dimensional arrangement of the nanotubes. Therefore, it would be interesting to investigate whether a similar effect could be achieved in an alcohol solvent other than EtOH. However, it is also possible, though unlikely, that the amorphous carbon was removed by EtOH and that the nanotubes were better aligned. TG comparison of the unchanged quality of the nanotubes before and after EtOH treatment would be a good way to determine whether EtOH treatment affected the behavior of the nanotubes. It would also be desirable to verify whether the EtOH treatment affects the behavior of the nanotubes by comparing the quality of the nanotubes before and after EtOH treatment with TG.

3 . ESI n-hexane typo.

The N in hexane must be corrected.

4 . Polyethyleneimine (PEI)

PEI, used as an n-dopant, has traditionally been known to have a short n-life imparted to

CNTs. However, it is now suggested that the thermoelectric performance of CNTs (including n-lifetime) varies depending on the molecular weight used and the amount contained in the film. On the other hand, in this study, the authors report that the n-lifetime with PEI doping is successful on a time scale. For this paper to be favored by many readers in the future, it would be desirable to at least quantify the PEI content coating the CNTs by TG.

5 . Figure S8

(b) The S value in CNT-e/c-PEI films is questionable. S values are often absolute values. If the S value for the maximum oxidation state is around $60 \mu\text{V K}^{-1}$ (Fig.1(c)), then the negative S value is rarely greater than this value. If any preliminary studies have been done on this, it would be good to include them in the manuscript.

6 . Lack of accuracy of Doping method in ESI

For n-materials, the n-doping method strongly affects thermoelectric performance and n-life. However, since the description of this method is simplified, it may be difficult for the reader to reproduce it. The following details should be provided the size, thickness, and placement of the film used in the container, the shape and manufacturer of the container used, the heating method (whether a magnetic stirrer or oil bath was used), and the cooling method. In addition, it would be helpful to specify the change in thermoelectric performance between the front and back surfaces of the dope film in this case.

7 . Figure 3 (f) and (g)

With reference to the following paper, the author can calculate the theoretical output power (P) and voltage (V) values. It is possible to know if the material performance can be maximized in the device in this work. It is expected that this will only be considered for the main devices.

Journal of Materials Chemistry A, vol. 5, pp. 15631–15639, 2017

ACS Applied Materials & Interfaces, vol. 11, pp. 29320–29329, 2019

8 . Atmospheric and water stability

In addition to imparting atmospheric stability to CNTs, PEI can also impart water stability. These results are shown in Figure S9 of the following paper.

Energy Mater. Adv., 2022 (2022), Article 9854657

Is this the same in this case? If possible, a companion study of the atmospheric or water stability of the device would further the claims of this paper.

9 . Semiconductor and metallic composition of CNTs

If possible, it would be good to investigate the semiconductor and metal composition of the nanotubes used in this production method, although technical difficulties are expected.

Reviewer #2 (Remarks to the Author):

This study presents the fabrication of carbon nanotube (CNT) films by chemical vapor deposition, with the introduction of ethanol solvent to enhance orientation via capillary effects. The resulting films exhibited a remarkably high p-type power factor, with the authors' attempts to dope the film with PEI suggesting the potential for achieving a high-performance n-type material.

While the experimental methodology appears robust, this study lacks novelty and unexpected results. Furthermore, there seems to be a potential misunderstanding of certain

concepts regarding the use of CNTs in thermoelectric applications. This does not meet the standards expected by journals such as Nat. Communications.

1. The observed high power factor is likely due to enhanced conductivity. Although the authors use a semi-empirical model based on recent research by Snyder's group to calculate mobility, it oversimplifies the explanation by attributing conductivity changes solely to mobility enhancement. In particular, variations in film density could alter the carrier density, which needs to be acknowledged by the authors.

2. While the emphasis on power factor is noteworthy, the omission of discussion of ZT (figure of merit) is concerning. Despite a high power factor, the ZT of the material remains significantly lower than benchmark materials, warranting a full discussion.

3. The discussion about cooling mechanisms seems confusing. If my interpretation is correct, the authors aim to use the material as a heat sink rather than a thermoelectric cooler. In such a context, the emphasis on power factors becomes less relevant, and the high thermal conductivity becomes the most important feature.

On closer examination, it appears to be scientifically uninteresting to classify this material as a thermoelectric material only. Instead, the focus should shift to discussions of electrical conductivity and transport mechanisms, similar to the approach taken in the referenced paper (DOI: 10.1038/ncomms3202).

Reviewer #3 (Remarks to the Author):

The manuscript entitled "Capillary compression induced outstanding n-type thermoelectric power factor in CNT films towards intelligent temperature controller" presents an experimental work on the alignment of CNTs for improved thermoelectric performance. This has resulted in significant improvements in n-type thermoelectric characteristics. Nevertheless, it is interesting to discuss the impact of each component on the observed characteristics. Authors may be eligible for publication after minor revision regarding the issues provided below.

1. In addition to securing flexibility, please mention in Introduction why a sample in the form of a film is needed in the thermoelectric field section. In addition, it is necessary to explain how TEG is formed and applied from film-type samples.

2. The English grammar is poor throughout this manuscript. The wrong verb tense is often used and, in many cases, the entire sentence is incomprehensible and confusing. A native English speaker needs to thoroughly edit this manuscript because the language is very distracting and reduces the impact of this work. There are also some misspelled words.

3. In some places, the authors used the word, "better", to compare the physical or mechanical properties, which needs to be rephrased.

4. Have the authors tried to measure the out-of-plane thermal conductivity of the composites? Have the authors made any attempts to determine if their films are anisotropic or isotropic? It is possible that the transport properties are anisotropic, with higher in-plane conductivities compared to transverse conductivities. High levels of anisotropy could help explain the unusual combination of properties exhibited by these films. Any comments about this would be helpful.

5. Have the authors measured the sheet resistance, depending on the systems? I assume that the electrical conductivity was obtained by calculating the sheet resistance (from the van der Pauw method) and film thickness (from SEM). In other words, the sheet resistance data need to be shown in the manuscript.

6. Although the authors explain a simultaneous increase in the composites using Hall Effect measurement (for carrier mobility), their assertion on decoupled thermoelectric behavior could be supported by using other analyses. I would suggest the authors to try to investigate electronic behaviors of the composites by characterizing cyclic voltammetry, Kelvin Probe Measurement, UV-vis NIR or UPS. These measurements would give the authors more detailed information to explain their assertion.

7. Band diagram of the individual constituent in the CNT-based films for p- and n-type systems should be provided, which is merit to explain energy-filtering effects at the interfaces among individual CNT.

8. In an energy filtering model, to explain the decoupled behavior between Seebeck coefficient and electrical conductivity, the energy potential barrier should exist at the interfaces in the CNT films, and the Seebeck coefficient value depends on the height of potential barrier. Do the authors have a specific model for energy diagram of potential barrier formed in the films from the work function and electron affinity? In an energy filtering model, the energy potential barrier also decreases the electrical conductivity from the highest value for free of energy potential barrier or each component. This should be considered in the proposed model to explain the simultaneous increase of Seebeck coefficient and electrical conductivity.

9. Are there carrier mobility or carrier concentration increase/decrease after PEI treatment? It should be analyzed using Hall Effect measurements.

Manuscript ID: NCOMMS-24-09055+

Title: Capillary compression induced outstanding n-type thermoelectric power factor in CNT films towards intelligent temperature controller

Reviewer Comments to Author:

Reviewer 1

General comments:

This paper consistently examines the manufacturing method of CNTs for thermoelectric applications, the anisotropic physical properties of p- and n-materials, and the evaluation of curl-type devices. I feel that this is an important insight that will open up previously unexplored possibilities of nanotube structures. Many experiments were performed, detailed manuscripts were written, and several interesting phenomena were discovered. Since the majority of fossil fuels are converted into unused heat, designing highly efficient power generation materials for energy harvesting is an eternal challenge and one of the important issues facing each country. Considering the combination of AI and IoT and the future development of IT technology in recent years, the approach discovered by the authors is very timely, and the content of the research results is also important. This paper is of great interest to readers, as this result is expected to contribute not only to the fields of engineering and materials, but also to the development of business in society and people's life work in the future. I generally agree with the arguments in this manuscript and consider its content to be of a high standard and appropriate for the journal. My small doubts and questions are summarized below. Although this manuscript is already interesting and informative, I will recommend it to this journal once these concerns and issues are addressed.

Authors' response:

Thank you very much for your comments. We truly appreciate your time and efforts.

Reviewer's Comments 1:

Change in film thickness due to pressing: I expect to find out more about the extent to which the film thickness changed before and after Scheme 1 compressing. Notably, the

prepared film had a thickness > 100 nm, and its TE performance was observed to be independent of the thickness of the CNT film (Energy Environ. Sci., 2017, 10, 2168.). I think this information is necessary for the reader to perform reproducible experiments. In Figure S6, it seems that the thickness of the sample was measured at only one location by SEM. It is more accurate to measure the film thickness of the sample at 200 locations using a film thickness meter instead of using an SEM and check the average value. I think it would be better to include at least the measured data of the main samples in this paper (CNT-o/c films, CNT-e/c films, CNT-o/c-PEI films, CNT-e/c-PEI films) in the ESI.

Authors' response:

Thank you very much for your good suggestions. The key factor that affects the TE performance of the CNT film is the porosity other than the film thickness. Detailed analysis has been provided in the revised supporting information. The paper (Energy Environ. Sci. 2017, 10, 2168) mentioned by the reviewer has been cited and the thickness of the CNT films before and after compressing have been provided to help the readers reproduce the experiments.

It would be true in the paper (Energy Environ. Sci. 2017, 10, 2168) that the TE performance was independent of the thickness of the CNT films because the packing density of CNTs in the films was almost the same. However, in this work, the packing density of CNTs in the films before compressing was lower than that of the CNTs in the films after compressing, which subsequently resulted in lower electrical conductivity of the CNT films before compressing than that of the CNT films after compressing. The electrical conductivity changes of the CNT films before and after compressing could be well understood with Maxwell-Eucken's equation as shown below:

$$\sigma_P = \sigma_0 \times \frac{1-P}{1+\beta P} \quad (1)$$

where P is the porosity, σ_P is the total electrical conductivity of a porous material, σ_0 is the intrinsic electrical conductivity of the materials, and β is the constant number determined by the conditions of the pores. The value of β is between 1.0 and 3.0 when the shape of the pores is almost the spherical style (J. Appl. Phys. 2006, 100, 044912; J. Alloys Compd. 2007, 432, 7; Thermochim. Acta 2015, 617, 83.). When the CNT films were compressed, the porosity decreased (the

packing density increased). Therefore, the electrical conductivity of the compressed CNT films was higher than that of the CNT films before compressing. Similar results have also been reported in previous works that the electrical conductivity is inversely proportional to the thickness (proportional to the density) of CNT films or their composite films after compressing (**Nat. Nanotechnol.** 2010, 5, 853; **Nat. Commun.** 2014, 5, 3848; **Nano Energy** 2021, 80, 105553; **Adv. Mater.** 2021, 33, 2103633; **Appl. Phys. Lett.** 2020, 116, 081902; **Appl. Phys. Lett.** 2009, 94, 012904). In the meanwhile, the compressing process was a physical process, which would not change the chemical environment of a single CNT, thus maintaining the Seebeck coefficient of the CNT films as suggested in previous works (**Adv. Mater.** 2024, 36, 2312570; **Adv. Funct. Mater.** 2022, 32, 2203080; **Small** 2023, 19, 2304266; **J. Mater. Chem. A** 2022, 10, 25740). The increased electrical conductivity and maintained Seebeck coefficient then resulted in the increase of thermoelectric power factor of CNT films after compressing in **Figure 1d** in the original manuscript. To conclude, it is demonstrated that the key factor affecting the TE performance of CNT films is the packing density (porosity) of the CNT film. Generally, the TE performance increases with the packing density of a CNT film. In this work, the thicknesses of the CNT film before compressing are in the range of $12.67 \pm 0.25 \mu\text{m}$ and $5.73 \pm 0.21 \mu\text{m}$ for CNT-o films and CNT-e films, respectively. Good reproducibility has been demonstrated in this work as well as in previously reported works (**Nat. Commun.** 2023, 14, 380; **Adv. Mater.** 2024, 36, 2312570; **Adv. Funct. Mater.** 2022, 32, 2203080; **Small** 2023, 19, 2304266; **J. Mater. Chem. A** 2022, 10, 25740) that significant improvement of TE performance could be observed while compressing a CNT film.

To help the readers to reproduce the experiments, the paper mentioned by the reviewer has been cited and related discussion has been added in the revised manuscript and the revised supporting information.

“During the compressing process, the key factor affecting the TE performance is the change of the porosity of the CNT films which has been well-discussed in previous works (**Adv. Mater.** 2024, 36, 2312570; **Adv. Funct. Mater.** 2022, 32, 2203080; **Small** 2023, 19, 2304266; **J. Mater. Chem. A** 2022, 10, 25740). A detailed analysis has been provided in the supporting information

(Figure S7), where the thickness of the CNT films decreased with the compressing time. For the CNT films with similar porosity, the TE performance will be independent of the thickness as demonstrated by Blackburn and Ferguson *et al.* (**Energy Environ. Sci.** 2017, 10, 2168). The thicknesses of the CNT film before compressing in this work were in the range of $12.67 \pm 0.25 \mu\text{m}$ and $5.73 \pm 0.21 \mu\text{m}$ for CNT-o films and CNT-e films, which were in the range of $1.12 \pm 0.05 \mu\text{m}$ and $0.91 \pm 0.03 \mu\text{m}$ after compressing”.

“The increase of the electrical conductivity of CNT films was due to the decrease of the porosity. SEM images in **Figure S5** indicated that the CNT films became dense after being compressed. The packing density of CNTs in the films before compressing was lower than that of the CNTs in the films after compressing (**Figure 1a, 1b** and **Figure S5**), which subsequently resulted in lower electrical conductivity of the CNT films before compressing than that of the CNT films after compressing (**Figure 1c**). The electrical conductivity changes of the CNT films before and after compressing could be well understood with Maxwell-Eucken's equation as shown below:

$$\sigma_P = \sigma_0 \times \frac{1-P}{1+\beta P} \quad (1)$$

where P is the porosity, σ_P is the total electrical conductivity of a porous material, σ_0 is the intrinsic electrical conductivity of the materials, and β is the constant number determined by the conditions of the pores. The value of β is between 1.0 and 3.0 when the shape of the pores is almost the spherical style (**J. Appl. Phys.** 2006, 100, 044912; **J. Alloys Compd.** 2007, 432, 7; **Thermochim. Acta** 2015, 617, 83.). When the CNT films were compressed, the porosity decreased (the packing density increased). The well alignment and dense packing of CNT resulted in the improvement of the carrier mobility in the films. Therefore, the electrical conductivity of the compressed CNT films was higher than that of the CNT films before compressing. Similar results have also been reported in previous works that the electrical conductivity is inversely proportional to the thickness (proportional to the density) of CNT films or their composite films after compressing (**Nat. Nanotechnol.** 2010, 5, 853; **Nat. Commun.** 2014, 5, 3848; **Nano Energy** 2021, 80, 105553; **Adv. Mater.** 2021, 33, 2103633; **Appl. Phys. Lett.** 2020, 116, 081902; **Appl. Phys. Lett.** 2009, 94, 012904). In the meanwhile,

the compressing process was a physical process, which would not change the chemical environment of a single CNT. Ultraviolet Photoelectron Spectroscopy (UPS) was performed with gold as a reference to identify the work function change of CNT-e and CNT-e/c films (**Figure S8**). The obtained work function of CNT-e film and CNT-e/c film exhibited similar values, indicating that the chemical environment of the CNT remained constant (the Fermi energy level is unchanged) before and after compressing. In addition, the pressure used in this work was only ~100 MPa which was far behind the pressure required to change the cylinder structure of a single CNT (several GPa, **Nat. Electron.** 2021, 4, 653; **J. Raman Spectrosc.** 2003, 34, 611; **Sci. Rep.** 2013, 3, 1331) Therefore, the Seebeck coefficient of the CNT films changed little after compressing. (**Adv. Mater.** 2024, 36, 2312570; **Adv. Funct. Mater.** 2022, 32, 2203080; **Small** 2023, 19, 2304266; **J. Mater. Chem. A** 2022, 10, 25740). The increased electrical conductivity and maintained Seebeck coefficient resulted in the increase of thermoelectric power factor of CNT films after compressing in **Figure 1d** in the original manuscript.”

Thanks to the reviewer for pointing this out. In **Figure S9** (**Figure S6** in the original manuscript), the film thickness was determined by taking the average distance of two lines other than checking the distance between two points. In the meanwhile, more than 3 samples were tested to minimize the errors. The achieved thicknesses should have been very close to the real thicknesses of the CNT films. The lines have been added in the TEM images in **Figure S9** to avoid the misunderstanding of readers. This method has been commonly used to determine the thickness of thin films in previous works (**Nat. Commun.** 2024, 15, 3426; **Nat. Commun.** 2023, 14, 380; **Adv. Mater.** 2023, 35, 2304751; **ACS Energy Lett.** 2021, 6, 4355; **Adv. Funct. Mater.** 2022, 32, 2203080; **Small** 2023, 19, 2304266; **J. Mater. Chem. A** 2022, 10, 25740). As suggested by the reviewer, the SEM images of the cross-section of CNT-o/c films, CNT-e/c films, CNT-o/c-PEI films, CNT-e/c-PEI films have been added in **Figure S9** in the revised supporting information (shown below).

Figure S9. The SEM images of CNT-o films (a), CNT-e films (b), CNT-o/c films (c), CNT-e/c films (d), CNT-o/c-PEI films (e), CNT-e/c-PEI films (f).

Reviewer's Comments 2:

EtOH treatment: I believe the author is making the claim that the EtOH treatment in Scheme 1 results in a one-dimensional arrangement of the nanotubes. Therefore, it would be interesting to investigate whether a similar effect could be achieved in an alcohol solvent other than EtOH. However, it is also possible, though unlikely, that the amorphous carbon was removed by EtOH and that the nanotubes were better aligned. TG comparison of the unchanged quality of the nanotubes before and after EtOH treatment would be a good way to determine whether EtOH treatment affected the behavior of the nanotubes. It would also be desirable to verify whether the EtOH treatment affects the behavior of the nanotubes by comparing the quality of the nanotubes before and after EtOH treatment with TG.

Authors' response:

Thank you very much for your time and efforts.

We agree with the reviewer that other alcohol solvents may also show a similar effect on the arrangement of CNTs. In previous works, various solvents have been studied to prepare well-arranged CNT fibers (**Nat. Commun.** 2021, 12, 4931; **Nat. Commun.** 2019, 10, 2962), CNT liquid crystals (**Science** 2013, 339, 182; **Sci. Adv.**, 2022, 8, eabm3285) and CNT films (post-treatment, **Nat. Commun.** 2023, 14, 380; **ACS Nano** 2020, 14, 14134). Different from the previous solution process and post-treatment method, we synthesized the well-arranged CNT films directly by combining the solvent effect and a floating catalyst chemical vapor deposition method. The newly developed process could produce CNT films at large scales, which could shorten the preparation process of high-quality CNT films as well. The environment-friendly solvent EtOH was used in this work. The obtained CNT films exhibited high TE performance as well. The main focus of the work is the new method for high-quality CNT films, improved TE performance of CNT films as well as the new concept of an intelligent temperature controller (ITC) with an LED alarm for potential applications in the thermal management of electronic devices. Although the study of the solvent effects with other solvents is very interesting, it may be out of the research scope of this work. Related work is undergoing in our lab, which will be published later.

The reviewer is right that it is unlikely that the EtOH could remove the amorphous carbon. Raman spectra showed that no significant change was observed in terms of the quality of CNT films with and without EtOH treatment. In the original manuscript, Raman spectroscopy was performed to identify the quality of the synthesized CNT films with/without EtOH treatment (**Figure S2**). Generally, two peaks are appearing at $\sim 1580\text{ cm}^{-1}$ (G-band) and $\sim 1350\text{ cm}^{-1}$ (D-band) and for CNT films, which are assigned to the graphitic carbon and the amorphous carbon (**Nat. Commun.** 2023, 14, 380; **Mater. Chem. Phys.** 2006, 96, 253; **Surf. Coat. Technol.** 1992, 50, 185; **J. Appl. Phys.** 2000, 88, 2305), respectively. **Figure S2** in the revised supporting information showed that the ratio between the intensity of G-band and D-band (I_G/I_D) were 18.25 and 21.64 for the CNT films with EtOH treatment (CNT-e) and the CNT films without EtOH treatment (CNT-o). The high I_G/I_D values showed that the quality of both the CNT-e and the CNT-o were better than typical CNTs which often exhibited a low

I_G/I_D in the range of 2-5 (**Carbon** 2017, 122, 496; **Nano Lett.** 2016, 16, 946; **Nat. Commun.** 2014, 5, 3848). The slightly lower I_G/I_D value indicated that CNT-e contained more amorphous carbon or defects than CNT-o, which demonstrated that EtOH would not remove the amorphous carbon in the film. Instead, it increased the defects in the CNT films, which might be due to the slightly increased oxidation of CNTs under wet conditions.

Additional experiments have been performed. The TGA curves of CNT-o films and CNT-e films are shown in **Figure S3**. CNT-o films and CNT-e films showed similar change trends and residual masses, indicating that ethanol treatment did not have a significant effect on CNT quality. The amorphous carbon content of CNT-o and CNT-e films was roughly calculated to be 5.5% and 8.1%, respectively, based on the weight change rate within 350-450 °C, as shown in **Figure S3** (**Nat. Commun.** 2023, 14, 380; **Carbon** 2009, 47, 3271; **Carbon** 2019, 144, 301; **Carbon** 2015, 88, 60). This is also consistent with the results of the Raman spectra shown in **Figure S2** that the I_G/I_D of CNT-e is slightly lower than that of CNT-o.

The related discussion has been added in the revised manuscript and the revised supporting information.

Figure S3. The TGA curve of CNT-o films and CNT-e films.

Reviewer's Comments 3:

ESI n-hexane typo: The N in hexane must be corrected.

Authors' response:

We are sorry for the typos. The “N-hexane” has been replaced with “n-hexane”.

Reviewer's Comments 4:

Polyethyleneimine (PEI): PEI, used as an n-dopant, has traditionally been known to have a short n-life imparted to CNTs. However, it is now suggested that the thermoelectric performance of CNTs (including n-lifetime) varies depending on the molecular weight used and the amount contained in the film. On the other hand, in this study, the authors report that the n-lifetime with PEI doping is successful on a time scale. For this paper to be favored by many readers in the future, it would be desirable to at least quantify the PEI content coating the CNTs by TG.

Authors' response:

Thank you very much for your good suggestions. Additional experiments have been performed. The TGA curve of CNT-e/c-PEI films is shown in **Figure S14**. The content of PEI in the CNT film was roughly calculated to be about 10.3% according to the weight variation of CNT-e/c-PEI films between 158-350 °C (**J. Power Sources** 2012, 210, 122; **Front. Energy Res.** 2020, 8, 196; **J. Environ. Chem. Eng.** 2019, 7, 103285; **Microporous Mesoporous Mater.** 2019, 275, 122).

The related discussion has been added in the revised manuscript and the revised supporting information.

Figure S14. The TGA curve of CNT-e/c films and CNT-e/c-PEI films.

Reviewer's Comments 5:

Figure S8 (b): The S value in CNT-e/c-PEI films is questionable. S values are often absolute values. If the S value for the maximum oxidation state is around $60 \mu\text{V K}^{-1}$ (Fig. 1(c)), then the negative S value is rarely greater than this value. If any preliminary studies have been done on this, it would be good to include them in the manuscript.

Authors' response:

Thank you very much for pointing this out. The description in the original manuscript of **Figure S15 (b)** (**Figure S8** changed to **Figure S15** in the revised manuscript) was correct. "The TE properties of PEI doped CNT films in the perpendicular direction have also been characterized (**Figure S15**). The maximum σ_{\perp} values are in the order of $\sigma_{\perp}(\text{CNT-o/c-PEI}) > \sigma_{\perp}(\text{CNT-e/c-PEI})$. The maximum PF_{\perp} values are in the order of $PF_{\perp}(\text{CNT-o/c-PEI}) > PF_{\perp}(\text{CNT-e/c-PEI})$." However, we used the wrong colors for the data points by mistake. We apologize for the confusion caused by the color misuse of the data points in **Figure S15 (b)**. The correct **Figure S15 (b)** is shown below and the wrong **Figure S15 (b)** has been replaced with the correct one.

Figure S15(b). The TE properties in the perpendicular direction of CNT-e/c-PEI films (b) as a function of vapor treatment time.

Reviewer's Comments 6:

Lack of accuracy of Doping method in ESI: For n-materials, the n-doping method strongly affects thermoelectric performance and n-life. However, since the description of this method is simplified, it may be difficult for the reader to reproduce it. The following details should be provided the size, thickness, and placement of the film used in the container, the shape and manufacturer of the container used, the heating method (whether a magnetic stirrer or oil bath was used), and the cooling method. In addition,

it would be helpful to specify the change in thermoelectric performance between the front and back surfaces of the dope film in this case.

Authors' response:

Thank you very much for your good suggestion. Detailed doping procedure has been provided in the revised supporting information as shown below:

“The vapor doping was performed by attaching the CNT films (10 mm×30 mm×0.9 μm) on the upper side of a glass-made cylindrical sublimation chamber (40 mm diameter, 70 mm height, Yangzhou Sunflower Glass Instrument Factory). At the same time, a certain amount (0.2 g) of n-type dopants (PEI, Me-TBD, TBD or TMG) was placed on the bottom of the cylindrical sublimation chamber. The chamber was heated on a hot plate with a sand bath and without a magnetic stirrer at 120 °C under ambient pressure. Therefore, dopant molecules are deposited on the CNT film surface after a certain time of heating (from 10 min to 120 min, **Figure 2, Figure S15 and Figure S19**). After finishing doping, the samples were left to cool down to room temperature naturally in the air. The scheme of the vapor doping process is shown in **Figure a.**”

Figure a. Schematic of the vapor doping process.

Additional experiments have been performed to specify the change in thermoelectric performance between the front and back surfaces of the PEI doped CNT films. The electrical conductivity, the Seebeck coefficient and the PF of the front and back sides of the CNT-e/c-PEI films were almost consistent as shown in **Figure S13**. The results indicated that the n-type CNT films were uniformly doped by the PEI vapor.

Related contents have been added in the revised manuscript and the revised supporting information.

Figure S13. The thermoelectric performance between the front and back surfaces of CNT-e/c-PEI films.

Reviewer's Comments 7:

Figure 3 (f) and (g) With reference to the following paper, the author can calculate the theoretical output power (P) and voltage (V) values. It is possible to know if the material performance can be maximized in the device in this work. It is expected that this will only be considered for the main devices.

Journal of Materials Chemistry A, vol. 5, pp. 15631–15639, 2017

ACS Applied Materials & Interfaces, vol. 11, pp. 29320–29329, 2019

Authors' response:

Thank you very much for your good suggestion. The theoretical open-circuit voltage and output power have been calculated according to the literature (**J Mater. Chem. A** 2017, 5, 15631; **ACS Appl. Mater. Interfaces** 2019, 11, 29320.)

The theoretical open-circuit voltage (V_{TH}) was calculated from the equation of $V_{TH} = N(|S_p| + |S_n|)\Delta T$, where N was the number of p–n modules and S_p and S_n were the Seebeck coefficients of p-type and n-type TE materials, respectively. When the temperature difference was 85 K, the maximum experimental open-circuit voltage (V_{EX}) was 81.5 mV, which was lower than V_{TH} (103 mV, **Figure S25**).

Figure S25. The open-circuit voltage in different temperature differences of SP-TEG/par.

The theoretical output power (P_{TH}) was calculated from the equation of $P_{TH} = \frac{V_{th}^2}{4R_{in}}$, where R_{in} was the resistance of TEG. When R_{in} was 56 Ω and the temperature difference was 85 K, the P_{TH} was 47.4 μW , which was larger than the experimental output power (29.7 μW) of SP-TEG/par. The output power of SP-TEG/par measured experimentally was only 60.1 % of the theoretical output power. The reason for experimental output power being lower than the theoretical values can be attributed to the fact that the poor heat dissipation at the cold side results in the higher temperature of the cold side compared to the ambient temperature, thus leading to a lower temperature difference between the hot side and the cold side of the TEG. The same issue has also been reported in the previous literature with low experiment to theory values (58.4 %, **ACS Energy Letter**, 2021, 6, 4355; 0.2%, **Energy Environ. Sci.**, 2014, 7, 1959; 54.1%, **Adv. Sci.** 2021,8, 2004947).

Related contents have been added in the revised manuscript and supporting information.

Reviewer's Comments 8:

Atmospheric and water stability: In addition to imparting atmospheric stability to CNTs, PEI can also impart water stability. These results are shown in Figure S9 of the following paper.

Energy Mater. Adv., 2022 (2022), Article 9854657

Is this the same in this case? If possible, a companion study of the atmospheric or water stability of the device would further the claims of this paper.

Authors' response:

Thank you very much for your good suggestion.

Additional experiments have been performed. The output power stability of the thermoelectric generator is shown in **Figure S27** (shown below). The output power of par-TEG showed good stability in the air at room temperature, which retained nearly 95% of its output power in 30 days. The output power of the par-TEG, which was stored in water, decreased with time. After 30 days, it maintained only 20% of the original value. While the par-TEG was encapsulated with polyethylene terephthalate (PET) films, it could maintain above 90% of the initial value after being kept under water for over 30 days. Although the prepared SP-TEG/par was not able to maintain 80% of the original output power after 30 days while being immersed in water directly as reported in the previous literature (**Energy Mater. Adv.** 2022, 2022, 9854657), the encapsulated SP-TEG/par exhibited superior stability of the output power in water.

Related contents have been added in the revised manuscript and supporting information.

Figure S27. The output power stability of par-TEG (in air, in water and in water after encapsulation).

Reviewer's Comments 9:

Semiconductor and metallic composition of CNTs: If possible, it would be good to investigate the semiconductor and metal composition of the nanotubes used in this production method, although technical difficulties are expected.

Authors' response:

Thank you very much for your time and efforts. The obtained CNT films typically exhibited metallic properties and it would be extremely hard to quantitatively know the semiconductor and metal composition of the nanotubes.

Additional experiments have been performed to identify the CNTs. High-resolution transmission electron microscopy (TEM) images showed that the majority of the CNTs were multi-walled carbon nanotubes (MWCNTs) as shown in **Figure S1**. Similar results have been reported in previous works that MWCNTs were obtained when the same method was used in the synthesis process (**Nat. Commun.**, 2023, 14, 380). As it is known that the band gap is inversely proportional to the diameter of CNTs, MWCNTs typically exhibit metal-like electrical properties (**J Mater. Chem. A** 2020, 8, 13095; **J Mater. Chem. A** 2021, 9, 3341). The high Seebeck coefficient of the CNT films can be attributed to the Fermi energy level being close to the 1D van Hove singularity of CNTs as suggested in the previous literature (**Nat. Commun.** 2021, 12, 4931; **Adv. Funct. Mater.** 2022, 32, 2203080; **Small** 2023, 19, 2304266; **J Mater. Chem. A** 2020, 8, 13095; **J Mater. Chem. A** 2021, 9, 3341).

Related contents have been added in the revised manuscript and supporting information.

Figure S1. The TEM images of CNT.

Reviewer 2

General comments:

This study presents the fabrication of carbon nanotube (CNT) films by chemical vapor deposition, with the introduction of ethanol solvent to enhance orientation via capillary effects. The resulting films exhibited a remarkably high p-type power factor, with the authors' attempts to dope the film with PEI suggesting the potential for achieving a high-performance n-type material. While the experimental methodology appears robust, this study lacks novelty and unexpected results. Furthermore, there seems to be a potential misunderstanding of certain concepts regarding the use of CNTs in thermoelectric applications. This does not meet the standards expected by journals such as Nat. Communications.

Authors' response:

Thank you very much for your time and efforts. We sincerely appreciate it.

Single carbon nanotube has remarkable electrical properties. However, these outstanding properties have remained elusive in macroscopic films (**Science** 2013, 339, 182). In previous works, various solvents have been studied to prepare well-arranged CNT fibers (**Nat. Commun.** 2021, 12, 4931; **Nat. Commun.** 2019, 10, 2962), CNT liquid crystals (**Science** 2013, 339, 182; **Sci. Adv.** 2022, 8, eabm3285) and CNT films (post-treatment, **Nat. Commun.** 2023, 14, 380; **ACS Nano** 2020, 14, 14134). Different from the previous solution process and post-treatment method, we synthesized the well-arranged CNT films directly by combining the solvent effect and a floating catalyst chemical vapor deposition method. The newly developed process could produce CNT films at large scales, which could shorten the preparation process of high-quality CNT films as well. The environment-friendly solvent EtOH was used in this work. The obtained CNT films exhibited high p- and n-type TE performance as well. The main focus of the work is the new method for high-quality CNT films, improved TE performance of CNT films, especially for n-type CNT films, and high normalized power output density that is higher than that of previously reported flexible all-inorganic thermoelectric generators. A new concept of an intelligent temperature controller (ITC) with an LED alarm was intentionally proposed to take advantage of the high power factor and high thermal conductivity of CNT films for potential applications in the thermal management of electronic devices, which exhibited

automated temperature controlling ability. It was different from the use of CNTs in traditional thermoelectric cooling applications, which paved an unusual way for the thermoelectric applications of high-performance CNT films.

Related contents have been added in the introduction of the revised manuscript. We hope this clarification addresses your concerns regarding the novelty of our study. We appreciate your time and consideration, and we remain open to any further suggestions or feedback you may have.

Reviewer's Comments 1:

The observed high power factor is likely due to enhanced conductivity. Although the authors use a semi-empirical model based on recent research by Snyder's group to calculate mobility, it oversimplifies the explanation by attributing conductivity changes solely to mobility enhancement. In particular, variations in film density could alter the carrier density, which needs to be acknowledged by the authors.

Authors' response:

Thank you very much for your time and efforts. We sincerely appreciate it.

The observed high power factor is due to both the enhanced electrical conductivity and the maintained Seebeck coefficient of the CNT films. As claimed in the original manuscript, “the decoupling of the electrical conductivity and the Seebeck coefficient then leads to a high maximum p-type PF in the parallel direction of the CNT-e/c films ($PF_{//(\text{CNT-e/c})}$), up to $9.31 \text{ mW m}^{-1} \text{ K}^{-2}$ (**Figure 1d**).” Similar results have been reported in previous works that the power factor of CNT films was improved by the denitrification methods due to the enhanced electrical conductivity and the maintained Seebeck coefficient (**Nat. Commun.** 2021, 12, 4931; **Adv. Mater.** 2024, 36, 2312570; **Adv. Funct. Mater.** 2022, 32, 2203080; **Mater. Today Commun.** 2017, 10, 41; **ACS Appl. Energy Mater.** 2021, 4, 4081; **Small** 2023, 19, 2304266; **J Mater. Chem. A** 2020, 8, 13095; **J Mater. Chem. A** 2021, 9, 3341).

We agree with the reviewer that the volume of the films decreases which will lead to the increase of the carrier density according to the definition of the carrier density if considering the voids as one of the components of the CNT films. In this case, it will be hard to understand the variation in the electrical properties of

CNT films before and after compressing. Because the Seebeck coefficient changed little during the compressing process, which did not agree with the suggestion that “variations in film density could alter the carrier density”.

To understand the change in the electrical conductivity and the Seebeck coefficient of the CNT films during compressing, it would be better to focus on the “effective” conducting components which is the CNT in the films. It is believed that the change in carrier mobility is responsible for the high power factor rather than the density of states. The physical compressing process would change the packing structure of CNTs in the films, which had little effect on the chemical environment of the CNTs (doping/de-doping). In the meanwhile, pressure used in this work was only ~100 MPa which was far away behind the pressure required to change the cylinder structure of a single CNT (several GPa, *Nat. Electron.* 2021, 4, 653-663; *J. Raman Spectrosc.* 2003, 34, 611-627; *Sci. Rep.* 2013, 3, 1331). Therefore, the Seebeck coefficient of the CNT films changed little after compressing (**Figure 1c**). The electrical conductivity increased significantly due to the well alignment and dense packing of CNT which resulted from the improvement of the carrier mobility in the films. This mechanism could be extended to many other composite systems that are comprised of highly conductive moieties and non-conductive moieties (*Adv. Mater.* 2024, 36, 2312570; *Adv. Funct. Mater.* 2022, 32, 2203080; *Mater. Today Commun.* 2017, 10, 41; *ACS Appl. Energy Mater.* 2021, 4, 4081; *Small* 2023, 19, 2304266; *J Mater. Chem. A* 2020, 8, 13095; *J Mater. Chem. A* 2021, 9, 3341). A similar mechanism has also been proposed by Chung, Park and Kim *et. al.* (*Adv. Energy Mater.* 2022, 12, 2200256).

Related content has been added to the revised manuscript and supporting information.

Reviewer's Comments 2:

While the emphasis on power factor is noteworthy, the omission of discussion of ZT (figure of merit) is concerning. Despite a high power factor, the ZT of the material remains significantly lower than benchmark materials, warranting a full discussion.

Authors' response:

Thank you very much for your time and efforts. The thermal conductivities of

CNT-e/c films ($114.3 \text{ W m}^{-1} \text{ K}^{-1}$) and CNT-e/c-PEI films ($86.23 \text{ W m}^{-1} \text{ K}^{-1}$) were provided in the original manuscript. ZT values of these CNT films have been calculated and added in the revised manuscript.

It was suggested in previous works that the primary concern was not to pursue high ZT for high energy efficiency but to generate enough electric power to support the electronic devices in many practical waste heat recovery applications when the waste heat was abundant and released freely (**Nat. Commun.** 2021, 12, 4931; **Adv. Energy Mater.** 2021, 11, 2100580; **Adv. Funct. Mater.** 2022, 32, 2203080; **Small** 2023, 19, 2304266). In these scenarios, the output power density became the key factor for thermoelectric devices, which was proportional to the power factor of thermoelectric materials. The temperature difference could be maintained by optimizing the length of the thermoelectric legs, especially for organic thermoelectric materials which were easy to process. As suggested in previous literature, the temperature decreased fast along the thin-film thermoelectric materials with high thermal conductivity (**J. Mater. Chem. A** 2022, 10, 25740). Therefore, the high power factor CNT films would be promising for making high power output density thermoelectric generators. The normalized power output density of a CNT film based flexible thermoelectric generator was comparable to/even higher than that of inorganic materials based flexible thermoelectric generators (**Figure 3i** in the original manuscript).

In the meanwhile, the ZT value increased by 1.2-2.5 times for the CNT films after compressing and vapor doping as shown in **Figure S18**. The reason for the increase of ZT was attributed to the increase of the electrical conductivity/thermal conductivity ratio since the compressed CNT films had different scattering effects on electrons and phonons due to their different mean free paths. Similar results have also been obtained in previous works (**Adv. Funct. Mater.** 2022, 32, 2203080; **J. Mater. Chem. A** 2022, 10, 25740; **Small** 2023, 19, 2304266).

Related content has been added to the revised manuscript and supporting information.

Figure S18. The ZT value of CNT films.

Reviewer's Comments 3:

The discussion about cooling mechanisms seems confusing. If my interpretation is correct, the authors aim to use the material as a heat sink rather than a thermoelectric cooler. In such a context, the emphasis on power factors becomes less relevant, and the high thermal conductivity becomes the most important feature. On closer examination, it appears to be scientifically uninteresting to classify this material as a thermoelectric material only. Instead, the focus should shift to discussions of electrical conductivity and transport mechanisms, similar to the approach taken in the referenced paper (DOI: 10.1038/ncomms3202).

Authors' response:

Thank you very much for your time and efforts. We used the materials as both a heat sink for passive cooling at a low temperature to save energy and a temperature alarm at a high temperature to monitor the temperature of the electronic devices to avoid overheating. The high power output of the thermoelectric generator made of the CNT films was essential to provide good sensitivity of temperature for future automated temperature controlling of electronic devices.

In this work, we reported a newly developed process to produce CNT films at large scales, which could shorten the preparation process of high-quality CNT films as well. The environment-friendly solvent EtOH was used here. The obtained CNT films exhibited high p- and n-type power factors, which resulted in a CNT film based high power output density flexible thermoelectric generator

that was comparable to/even higher than that of an inorganic materials based state-of-the-art flexible thermoelectric generator. At last, a new concept of an intelligent temperature controller (ITC) with an LED alarm was intentionally proposed to take advantage of the high power factor and high thermal conductivity of CNT films for potential applications in the thermal management of electronic devices, which exhibited automated temperature controlling ability. The results indicated that CNT-based thermoelectric materials could combine the passive cooling and the temperature alarm for future thermal management of electronic devices. It was different with the use of CNTs in traditional active thermoelectric cooling applications, which paved an unusual way for the thermoelectric applications of high-performance CNT films. The focus on the discussion of electrical conductivity and transport would be out of the research scope of this work.

Reviewer 3

General comments:

The manuscript entitled "Capillary compression induced outstanding n-type thermoelectric power factor in CNT films towards intelligent temperature controller" presents an experimental work on the alignment of CNTs for improved thermoelectric performance. This has resulted in significant improvements in n-type thermoelectric characteristics. Nevertheless, it is interesting to discuss the impact of each component on the observed characteristics. Authors may be eligible for publication after minor revision regarding the issues provided below.

Authors' response:

Thank you very much for your time and efforts. We sincerely appreciate it.

Reviewer's Comments 1:

In addition to securing flexibility, please mention in Introduction why a sample in the form of a film is needed in the thermoelectric field section. In addition, it is necessary to explain how TEG is formed and applied from film-type samples.

Authors' response:

Thank you very much for your suggestion. The description of film TEGs has been added to the introduction as follows:

“The film TE generators (TEGs) are often used for the recovery of waste heat at low to medium temperatures (**Joule** 2019, 3, 53; **Nat. Nanotechnol** 2023, 18, 1281; **Nat. Commun.** 2023, 14, 380; **Nat. Commun.** 2023, 14, 8442), which have great potential for application in the field of wearable electronic devices due to their good flexibility as compared with pellet structure TEGs. The thin-film TEG usually prepared from organic thermoelectric materials also has the advantages of being lightweight, inexpensive, non-toxic and easily processed. Different from traditional film TEGs obtained by connecting p-type and n-type strip film through the connection way of "electrical series, thermal parallel" with conductive silver paste and metal electrodes (copper, silver, *etc*), a new type of single-piece film TEGs with lower contact resistance has been reported, which were prepared by patterned cutting after printing or vapor doping to create p-type and n-type thermoelectric legs in films (**Adv. Mater.** 2024, 36, 2312570; **ACS**

Energy Lett. 2021, 6, 4355; **Nano Energy** 2021, 93, 106789; **Small** 2023, 19, 2304266).”

Reviewer's Comments 2:

The English grammar is poor throughout this manuscript. The wrong verb tense is often used and, in many cases, the entire sentence is incomprehensible and confusing. A native English speaker needs to thoroughly edit this manuscript because the language is very distracting and reduces the impact of this work. There are also some misspelled words.

Authors' response:

Thank you very much for pointing this out. The manuscript has been checked throughout for grammatical and spelling errors, and all the corresponding errors should have been corrected.

Reviewer's Comments 3:

In some places, the authors used the word, "better", to compare the physical or mechanical properties, which needs to be rephrased.

Authors' response:

Thank you very much for your good suggestions. The relevant contents have been rephrased. The details are as follows:

“Theoretically, 1D materials such as carbon nanotubes (CNTs) even have a better PF of $100 \text{ mW m}^{-1} \text{ K}^{-2}$.” has been corrected to “Theoretically, 1D materials such as carbon nanotubes (CNTs) even have a higher PF of $100 \text{ mW m}^{-1} \text{ K}^{-2}$.”.

“Then, the CNT-e and CNT-o films were further compressed at a pressure of about 100 MPa to improve the packing density of CNTs for better electrical conductivity as suggested in literature.” has been rephrased as “Then, the CNT-e and CNT-o films were further compressed at a pressure of about 100 MPa to improve the packing density of CNTs for higher electrical conductivity as suggested in the literature.”.

“Ethanol treated CNT films (CNT-e) show better alignment than the CNT-o films obviously.” has been rephrased as “Ethanol treated CNT films (CNT-e) show higher alignment than the CNT-o films obviously.”.

“SEM images show that the compressed CNT-e films (CNT-e/c) have better

alignment than the compressed CNT-o films (CNT-e/c).” has been corrected to “SEM images show that the compressed CNT-e films (CNT-e/c) have higher alignment than the compressed CNT-o films (CNT-e/c).”.

Reviewer's Comments 4:

Have the authors tried to measure the out-of-plane thermal conductivity of the composites? Have the authors made any attempts to determine if their films are anisotropic or isotropic? It is possible that the transport properties are anisotropic, with higher in-plane conductivities compared to transverse conductivities. High levels of anisotropy could help explain the unusual combination of properties exhibited by these films. Any comments about this would be helpful.

Authors' response:

Thank you very much for your question. The attempts to detect the out-of-plane thermal conductivity of CNT films failed due to the ultra-thin thickness ($\sim 1 \mu\text{m}$) of the prepared carbon nanotube films, making their out-of-plane thermal conductivity difficult to measure. Similar results have been reported in the previous literature (**Nano Energy** 2022, 93, 106804; **J Mater. Chem. A** 2021, 9, 3341; **J Mater. Chem. A** 2020, 8,13095). The in-plane thermal conductivity in the direction perpendicular to the CNT orientation (k_{\perp}) would be similar to the out-of-plane thermal conductivity (k_{out}), due to the good alignment of the carbon nanotubes, as shown in **Figure S17**. The k_{\perp} values of CNT films are shown in **Figure S16b**. The thermal conductivity in the perpendicular direction was significantly lower than that in the parallel direction, which showed significant anisotropy.

Figure S17. Schematic of thermal conductivity in different directions of carbon nanotube films.

Figure S16. The thermal conductivity of CNT films.

CNT films were proved to be anisotropic by polarized Raman spectroscopy as well, as shown in **Figure S2**. The G-band intensity of CNT-e films in the direction parallel to the rolling direction ($I_{G//}$) was about 3.25 times higher than that in the perpendicular direction ($I_{G\perp}$), while the $I_{G//} : I_{G\perp}$ for CNT-o films was only 2.26. The results indicated that CNT films were anisotropic. In addition, the electrical conductivity and thermal conductivity of CNT films were also anisotropy, as shown in **Figure S7**, **Figure S15**, and **Figure S16**. The electrical conductivity and thermal conductivity of the CNT-e/c film in the parallel direction were 2.17 MS m^{-1} and $114.3 \text{ W m}^{-1} \text{ K}^{-1}$, which were 4.26 and 1.94 times higher than their perpendicular direction, respectively.

Related content has been added to the revised manuscript and supporting information.

Figure S2. The Raman spectra of CNT-o films and CNT-e films.

Reviewer's Comments 5:

Have the authors measured the sheet resistance, depending on the systems? I assume that the electrical conductivity was obtained by calculating the sheet resistance (from the van der Pauw method) and film thickness (from SEM). In other words, the sheet resistance data need to be shown in the manuscript.

Authors' response:

Thank you very much for your comments. The electrical conductivity of the film was measured by commercial equipment (NETZSCH, SBA-458, Germany) with a four-probe method. All measurements were carried out at room temperature under argon protection and the CNT film samples were cut into strips 20 mm long and 4 mm wide. The schematic of the anisotropic electrical conductivity measurements for CNT films is shown in **Figure S6**. The electrical conductivity (σ) of the samples was calculated by the equation: $\sigma = 1/\rho = L/RA$, where ρ was the resistivity, L was the length of the sample between the electrodes, R was the resistance and A was the cross-sectional area of the sample. This method has been widely used for the electrical conductivity measurement of films (**Nat. Mater.** 2019, 18, 62; **Nat. Commun.** 2023, 14, 380; **Adv. Mater.** 2024, 36, 2312570; **Energy Environ. Sci.** 2020, 13, 3480; **Adv. Funct. Mater.** 2022, 32, 2203080; **J. Mater. Chem. A** 2022, 10, 25740; **Small** 2023, 19, 2304266).

Additional experiments have been performed to verify the electrical conductivity of CNT films by the van der Pauw method. The sheet resistance of CNT-e/c films was measured according to previous literature (**Phys. Rev. B** 2004, 70, 165307; **Phys. Rev. Lett.** 1999, 83, 4223; **Nat., Phys.** 2024, DOI: 10.1038/s41567-024-02443-x; **Mater. Lett.** 2013, 105, 20; **ACS Nano** 2017, 15, 7226). The sheet resistance of CNT-e/c films in the parallel direction measured by the van der Pauw method was consistent with the sheet resistance calculated from the electrical conductivity measured by SBA-458, as shown in **Table S1**.

Related content has been added to the revised manuscript and supporting information.

Figure S6. The schematic of the anisotropic electrical conductivity measurements for CNT films.

Table S1. The sheet resistance of CNT-e/c films.

		σ (MS m ⁻¹)	Sheet resistance calculated by σ ($\Omega \square^{-1}$)	Sheet resistance measured by the van der Pauw method ($\Omega \square^{-1}$)
CNT-e/c film	//	2.17 ± 0.07	0.51 ± 0.03	0.5 ± 0.01

Reviewer's Comments 6:

Although the authors explain a simultaneous increase in the composites using Hall Effect measurement (for carrier mobility), their assertion on decoupled thermoelectric behavior could be supported by using other analyses. I would suggest the authors to try to investigate electronic behaviors of the composites by characterizing cyclic voltammetry, Kelvin Probe Measurement, UV-vis NIR or UPS. These measurements would give the authors more detailed information to explain their assertion.

Authors' response:

Thank you very much for your good suggestions. Additional experiments have been performed to explain the increased electrical conductivity and the maintained Seebeck coefficient after compression.

Ultraviolet Photoelectron Spectroscopy (UPS) was performed with gold as a reference to identify the work function change of CNT-e and CNT-e/c films (**Figure S8**). The work function was obtained by the equation: $\phi = h\nu + |E_F| - |E_{cutoff}|$, where ϕ was the work function, $h\nu$ was the incoming photon energy from the He I source of 21.2 eV, and $|E_F| - |E_{cutoff}|$ was the difference in energy between the onset of the secondary electrons and the Fermi edge. The obtained work function of CNT-e film and CNT-e/c film exhibited similar values, indicating that the chemical environment of the CNT remained constant (the Fermi energy level is unchanged) before and after compressing, which resulted

in the maintenance of the Seebeck coefficient.

Related contents have been added in the revised manuscript and supporting information.

Figure S8. UPS results of CNT-a and CNT-e/c films.

Reviewer's Comments 7:

Band diagram of the individual constituent in the CNT-based films for p- and n-type systems should be provided, which is merit to explain energy-filtering effects at the interfaces among individual CNT.

Authors' response:

Thank you very much for your suggestion.

The energy-filtering effect usually emerges from the barrier blockage of low-energy carriers by the potential barrier, resulting in a significant increase in the Seebeck coefficient, while the electrical conductivity remains constant (**Adv. Phys.** 2018, 67, 69; **Materials** 2017, 10, 418). The energy filtering effect was usually observed in carbon nanotube-organic/inorganic composites, as reported in previous literature (**Adv. Funct. Mater.** 2024, 2315677; **Nanoscale**, 2022, 14, 9419). However, this effect may not fulfill the scenario of CNT-only films in this work since the Seebeck coefficient of the CNT films maintained nearly constant before and after compressing (**Figure 1c**).

Ultraviolet Photoelectron Spectroscopy (UPS) was performed with gold as a reference to identify the work function change of CNT-e and CNT-e/c films (**Figure S8**). The work function was obtained by the equation: $\phi = h\nu + |E_F| - |E_{cutoff}|$, where ϕ was the work function, $h\nu$ was the incoming photon energy from the He I source of 21.2 eV, and $|E_F| - |E_{cutoff}|$ was the difference in energy between the onset of the secondary electrons and the Fermi edge. The obtained work function of CNT-e film and CNT-e/c film exhibited similar values, indicating that the chemical environment of the CNT remained constant (the

Fermi energy level is unchanged) before and after compressing, which resulted in the maintenance of the Seebeck coefficient.

Related content has been added to the revised manuscript and supporting information.

Figure S8. UPS results of CNT-a and CNT-e/c films.

Reviewer's Comments 8:

In an energy filtering model, to explain the decoupled behavior between Seebeck coefficient and electrical conductivity, the energy potential barrier should exist at the interfaces in the CNT films, and the Seebeck coefficient value depends on the height of potential barrier. Do the authors have a specific model for energy diagram of potential barrier formed in the films from the work function and electron affinity? In an energy filtering model, the energy potential barrier also decreases the electrical conductivity from the highest value for free of energy potential barrier or each component. This should be considered in the proposed model to explain the simultaneous increase of Seebeck coefficient and electrical conductivity.

Authors' response:

Thank you very much for your time and efforts. We sincerely appreciate it. The mechanism of decoupling the electrical conductivity and the Seebeck coefficient has been proposed in previous literature (**Adv. Mater.** 2024, 36, 2312570; **Adv. Funct. Mater.** 2022, 32, 2203080; **Mater. Today Commun.** 2017, 10, 41; **ACS Appl. Energy Mater.** 2021, 4, 4081; **Small** 2023, 19, 2304266; **J Mater. Chem. A** 2020, 8, 13095; **J Mater. Chem. A** 2021, 9, 3341; **Adv. Energy Mater.** 2022, 12, 2200256).

To understand the change of the electrical conductivity and the Seebeck coefficient of the CNT films during compressing, it would be better to focus on the “effective” conducting components which are CNT in the films. It is believed

that the change in carrier mobility is responsible for the high power factor rather than the density of states. The physical compressing process would change the packing structure of CNTs in the films, which had little effect on the chemical environment of the CNTs (doping/de-doping). In the meanwhile, pressure used in this work was only ~100 MPa which was far away behind the pressure required to change the cylinder structure of a single CNT (several GPa, **Nat. Electron.** 2021, 4, 653; **J. Raman Spectrosc.** 2003, 34, 611; **Sci. Rep.** 2013, 3, 1331). Therefore, the Seebeck coefficient of the CNT films changed little after compressing. The electrical conductivity increased significantly due to the well alignment and dense packing of CNT that resulted from the improvement of the carrier mobility in the films. This mechanism could be extended to many other composite systems that are comprised of highly conductive moieties and non-conductive moieties (**Adv. Mater.** 2024, 36, 2312570; **Adv. Funct. Mater.** 2022, 32, 2203080; **Mater. Today Commun.** 2017, 10, 41; **ACS Appl. Energy Mater.** 2021, 4, 4081; **Small** 2023, 19, 2304266; **J Mater. Chem. A** 2020, 8, 13095; **J Mater. Chem. A** 2021, 9, 3341). A similar mechanism has also been proposed by Chung, Park and Kim *et al.* (**Adv. Energy Mater.** 2022, 12, 2200256).

Related contents have been added in the supporting information.

Reviewer's Comments 9:

Are there carrier mobility or carrier concentration increase/decrease after PEI treatment?

It should be analyzed using Hall Effect measurements.

Authors' response:

Thank you very much for your suggestion. It is challenging to get the accurate value of carrier mobility for CNT films due to the quantum confinement effect as reported by Tanabe *et al.* in literature (**Nano Lett.** 2011, 11, 3190; **Phys. Today** 1988, 41, 36). In addition, the contained Fe nanoparticles made the Hall measurement data of the CNT films worse. Therefore, we used weighted mobility reported by Snyder's group to evaluate the variation of carrier mobility in CNT films, since the weighted mobility had a similar trend with the Hall mobility, as reported in the previous literature (**Adv. Mater.** 2020, 32, 2001537; **ACS Nano** 2022, 16, 78; **J. Mater. Chem. A** 2022, 10, 3698; **J. Materiomics** 2021, 7, 742). The weighted mobility of the obtained matched well with the previously reported

literature for CNT (**Adv. Energy Mater.** 2022, 12, 2200256).

The weighted mobility of CNT films after PEI treatment exhibited a slight decrease, as shown in **Figure S12**. The slightly reduced weighted mobility can be attributed to the introducing of non-conducting PEI molecules, as suggested in the previous literature (**Nano Energy** 2021, 84, 105902; **Small** 2023, 19, 2304266).

Related contents have been added in the revised manuscript and supporting information.

Figure S12. The weight mobility in the parallel direction ($\mu_{||}$) of CNT-o/c films, CNT-e/c films, CNT-o/c-PEI films and CNT-e/c-PEI films.

REVIEWER COMMENTS

Reviewer #1 (Remarks to the Author):

Thanks for your answers to my questions. All my questions have been answered. It is my hope that the authors' paper will be read by more people and will lead to further development of thermoelectric CNTs.

Reviewer #2 (Remarks to the Author):

I believe the authors have tried to revise the paper, but I still have several concerns:

Why don't the authors measure the change in film density before and after pressing? This would be a very simple experiment. If the density changes closely match the conductivity changes, it would suggest that everything is related to film density.

My main concern is that CNT films are not used here as thermoelectric materials, which changes the motivation (introduction) for this work. As a heat sink, we are only interested in thermal conductivity. As a temperature sensor, we are interested in the Seebeck coefficient, especially its stability over a wide temperature range, rather than the absolute value. The high power factor is not critical for the applications.

Even if we consider it as a thermoelectric material, it is good but not exceptional because the ZT value is not very high.

Reviewer #3 (Remarks to the Author):

The authors have diligently done additional work on the issues raised by the reviewer, and we believe that all of their responses are acceptable.

Manuscript ID: NCOMMS-24-09055A

Title: Capillary compression induced outstanding n-type thermoelectric power factor in CNT films towards intelligent temperature controller

Reviewer Comments to Author:

Reviewer 1

General comments:

Thanks for your answers to my questions. All my questions have been answered. It is my hope that the authors' paper will be read by more people and will lead to further development of thermoelectric CNTs.

Authors' response:

Thank you very much for helping us to improve the quality of this manuscript.

We sincerely appreciate your time and efforts.

Reviewer 2

General comments:

I believe the authors have tried to revise the paper, but I still have several concerns:

Authors' response:

Thank you very much for helping us to improve the quality of this manuscript. We revised the paper according to your previous valuable comments. We appreciate any further suggestions and comments.

Reviewer's Comments 1:

Why don't the authors measure the change in film density before and after pressing? This would be a very simple experiment. If the density changes closely match the conductivity changes, it would suggest that everything is related to film density.

Authors' response:

Thank you very much for your time and efforts. We sincerely appreciate it.

We agree with the reviewer that the film density is important to the electrical conductivity of the CNT films, which is also critical to the electrical and mechanical properties of CNT fibers (**Nat Commun.** 2021, 12, 4931; **Adv. Energy Mater.** 2022, 12, 2200256; **Nano Lett.** 2016, 16, 946; **Carbon** 2017, 123, 593; **Carbon** 2016, 99, 407; **ACS Appl. Electron. Mater.** 2024, 6, 3, 2039; **Carbon** 2018, 136, 409; **Carbon** 2024, 219, 118845). However, it is still challenging to have densely packed CNTs in CNT films as well as in CNT fibers, which makes the remarkable electrical properties remain elusive in macroscopic CNT films and fibers (**Science** 2013, 339, 182). Efforts have been placed on improving the density of CNT films and fibers previously (**Nat. Commun.** 2021, 12, 4931; **Nat. Commun.** 2019, 10, 2962), CNT liquid crystals (**Science** 2013, 339, 182; **Sci. Adv.** 2022, 8, eabm3285) and CNT films (post-treatment, **Nat. Commun.** 2023, 14, 380; **ACS Nano** 2020, 14, 14134). Different from the previous solution process and post-treatment method, we synthesized the well-arranged CNT films directly by combining the solvent effect and a floating catalyst chemical vapor deposition method. The newly developed process could produce CNT films at large scales with high thermoelectric power factor due to the improved density, which could shorten the preparation process of high-quality CNT films as well.

Additional experiments have been performed to identify the change in film density before and after pressing according to the reviewer's good suggestion. **Figure S8** showed that the density of the CNT-e film (before compressing) was about 0.8 g cm^{-3} , which increased to $\sim 2.1 \text{ g cm}^{-3}$ for the CNT-e/c film (after compressing). The results supported the claim that the electrical conductivity was strongly related to the density of the CNT films. Similar results on the density dependent electric properties of CNT films have been reported in previous works. (**Adv. Funct. Mater.** 2022, 32, 2203080; **Small** 2023, 19, 2304266)

Related contents have been added in the revised supporting information.

Figure S8. The density of the CNT-e films before and after compressing.

Reviewer's Comments 2:

My main concern is that CNT films are not used here as thermoelectric materials, which changes the motivation (introduction) for this work. As a heat sink, we are only interested in thermal conductivity. As a temperature sensor, we are interested in the Seebeck coefficient, especially its stability over a wide temperature range, rather than the absolute value. The high power factor is not critical for the applications.

Authors' response:

Thank you very much for your time and efforts.

We guess that the reviewer referred to the application of CNT films in an intelligent temperature controller (ITC) with an LED alarm. The CNT film-based TEG was used as a heat sink for passive cooling at a low temperature to save energy, which was different from traditional applications of TEGs in active thermoelectric cooling that required power supply all the time. It was also used

as a power generator to support an LED temperature alarm at a high temperature to avoid overheating. The high power factor of the CNT films was important for the TEG to generate sufficient output power over the limited contact area of the heat source to support the LEDs for alarming. The output power density of the TEG was proportional to the power factor of CNT films as suggested in the literature (**Nat. Nanotechnol.** 2023, 18, 1281; **Nat. Commun.** 2023, 14:8442; **Energy Environ. Sci.** 2020, 13, 1240; **Adv. Energy Mater.** 2021, 11, 2100580; **Small** 2023, 19, 2304266; **J. Mater. Chem. A** 2022, 10, 25740). This LED alarm in the ITC was self-powered, which was different from traditional temperature sensors. The characterization results of the TEG was shown in **Figure 3**.

The whole paper focused on developing high power factor CNT films for thermoelectric devices. The thermoelectric properties of CNT films have been shown in **Figure 1** and **Figure 2** in the original manuscript. Whole CNT film thermoelectric generators have been fabricated to demonstrate the heat-to-electricity conversion ability of the achieved thermoelectric materials as shown in **Figure 3**. At last, a new concept of an intelligent temperature controller (ITC) with an LED alarm was intentionally proposed to take advantage of the high power factor and high thermal conductivity of CNT films for potential applications in the thermal management of electronic devices, which exhibited automated temperature controlling ability (**Figure 4**).

We hope the reviewer could understand that it would be not possible to discuss every high property in devices with only one paper due to the limitation of the length. For example, Herrmann and Zheng *et. al.* reported a high mechanical property hydrogel for potential applications in artificial muscles and soft robotics without fabricating soft robotics (**Nat. Commun.**, 2024, 15, 249). Kono *et. al.* reported a high-performance CNT fiber with a giant power factor and a high thermal conductivity for potential applications in flexible TEG and thermoelectric cooling without fabricating thermoelectric cooling devices (**Nat. Commun.** 2021, 12, 4931). In this work, a new concept of an intelligent temperature controller (ITC) with an LED alarm was intentionally proposed, instead of the active thermoelectric cooling device, which would extend the application range of the CNT films with a high power factor and a high thermal conductivity in the thermal management of electronic devices. In addition, although these CNT films were promising for the application of thermoelectric

cooling (**Nat. Commun.** 2021, 12, 4931; **ACS Energy Lett.** 2021, 6, 12, 4355; **Appl. Phys. Lett.** 2023, 123, 243901; **Phys. Rev. Appl.** 2019, 11, 054008; **Appl. Phys. Lett.** 2015, 106, 203506), the active thermoelectric cooling device was not fabricated due to the limitation of the experimental conditions in our lab which required precise measurement of the temperature and vacuum environments for the accurate evaluation of the Peltier effect of the materials and devices (**Nat. Commun.**, 2018, 9, 3586). Further discussion on the heat sink and the temperature sensors would be out of the research scope of this work.

Related contents have been added in the revised manuscript.

Reviewer's Comments 3:

Even if we consider it as a thermoelectric material, it is good but not exceptional because the ZT value is not very high.

Authors' response:

Thank you very much for your time and efforts.

It is true that the ZT value of CNT based thermoelectric materials is much lower than that of inorganic materials at room temperature due to their intrinsic high thermal conductivities. However, the excellent electrical and mechanical properties of CNTs make them promising candidates for flexible thermoelectric materials. Research on improving the thermoelectric properties of CNT based materials is still very attractive recently (**Nat. Commun.**, 2024, 15, 3426; **Adv. Mater.**, 2024, 36, 2312570; **Renew sust Energy Rev.**, 2024, 199, 114496; **Nat. Commun.**, 2023, 14, 380; **Nat. Commun.**, 2021, 12, 4931).

In the meanwhile, previous literature suggests that ZT should not be the only parameter to evaluate a thermoelectric material. In the past years, effects have been placed on pursuing high power factor of materials for the high output power density of a TEG (**Nat. Commun.** 2021, 12, 4931; **Adv. Energy Mater.** 2021, 11, 2100580; **Adv. Funct. Mater.** 2022, 32, 2203080; **Small** 2023, 19, 2304266). Because, for many practical waste heat recovery applications when the waste heat was abundant and released freely, the primary concern was not to pursue high energy conversion efficiency with high ZT materials but to generate enough electric power to support the electronic devices (**Adv. Energy Mater.** 2021, 11, 2100580). In these scenarios, the output power density became the key factor for

thermoelectric devices, which was proportional to the power factor of thermoelectric materials. The temperature difference could be maintained by optimizing the length of the thermoelectric legs, especially for organic thermoelectric materials which were easy to process. As suggested in previous literature, the temperature decreased fast along the thin-film thermoelectric materials with high thermal conductivity (**J. Mater. Chem. A** 2022, 10, 25740). Therefore, the high power factor CNT films would be promising for making high power output density thermoelectric generators.

We agree with the reviewer that the ZT of the CNT films is good, which is comparable to the state-of-the-art reported CNT films as shown in **Table S5** (**Nat Commun.** 2023, 14, 380; **Nano Lett.** 2012, 12, 1307; **Adv. Funct. Mater.** 2022, 32, 2203080; **Energy Environ. Sci.** 2012 5, 5364; **Adv. Funct. Mater.** 2016, 26, 3021; **Nano Energy** 2022, 93, 106804; **Small** 2023, 19, 2304266; **ACS Appl. Energy Mater.** 2020, 3, 6929). Furthermore, the CNT films exhibited exceptional power factors, which were comparable to/even higher than that of state-of-the-art inorganic thermoelectric films at room temperature (**Figures 1e** and **2f** in the original manuscript). The normalized power output density of a CNT film based flexible thermoelectric generator was comparable to/even higher than that of inorganic materials based flexible thermoelectric generators (**Figure 3i** in the original manuscript).

Related contents have been added in the revised manuscript and supporting information.

Table S5. Comparison of in-plane ZT values of CNT films at room temperature.

Materials	PF (mW m ⁻¹ K ⁻²)	k (W m ⁻¹ K ⁻¹)	ZT	Ref.
CNT-e/c film	9.31	114.3	2.5×10 ⁻²	This work
CSA-MWCNT film	4.66	45.9	3.0×10 ⁻²	5
MWCNT film	7.25	99.4	2.2×10 ⁻²	10
MWCNT film	0.34	5.5	1.9×10 ⁻²	50
MWCNT film	~1.05	21.9	1.4×10 ⁻²	5
SWCNT film	2.2×10 ⁻⁵	9.8	7.0×10 ⁻³	51

SWCNT film	$\sim 1.1 \times 10^{-2}$	~ 1	3.3×10^{-3}	52
SWCNT-8022- CN6CP film	0.16	17.2	2.8×10^{-3}	53
Doped SWCNT film	2.3×10^{-5}	39	2.0×10^{-3}	51
MWCNT film	-	26.95	$\sim 6.5 \times 10^{-4}$	9

Reviewer 3

General comments:

The authors have diligently done additional work on the issues raised by the reviewer, and we believe that all of their responses are acceptable.

Authors' response:

Thank you very much for helping us to improve the quality of this manuscript.

We sincerely appreciate your time and efforts.

REVIEWERS' COMMENTS

Reviewer #2 (Remarks to the Author):

I appreciate that the authors have clearly provided the film density data. While other reviewers strongly recommend the publication of this paper and I respect their decision. Personally, I find that altering material density to achieve a high power factor is not scientifically compelling. This is because both electrical conductivity and thermal conductivity exhibit a linear relationship with material density, meaning that changes in density do not necessarily enhance the performance of thermoelectric materials. In fact, reducing the material density could potentially improve the module's power output by better matching the internal resistance with the interfacial resistance.

Reviewer Comments to Author:

Reviewer 2

General comments:

I appreciate that the authors have clearly provided the film density data. While other reviewers strongly recommend the publication of this paper and I respect their decision.

Personally, I find that altering material density to achieve a high power factor is not scientifically compelling. This is because both electrical conductivity and thermal conductivity exhibit a linear relationship with material density, meaning that changes in density do not necessarily enhance the performance of thermoelectric materials. In fact, reducing the material density could potentially improve the module's power output by better matching the internal resistance with the interfacial resistance.

Authors' response:

Thank you very much for your kindly support. We sincerely appreciate your time and efforts.

Electrical conductivity and thermal conductivity didn't exhibit a linear relationship with material density in this work, which resulted in the increase of both the power factor and the ZT of the CNT films. Related results have been discussed in the original manuscript show as follows:

*“In the meanwhile, the ZT value increased by 1.2-2.5 times for the CNT films after compressing and vapor doping as shown in Figure S19. The reason for the increase of ZT was attributed to the increase of the electrical conductivity/thermal conductivity ratio since the compressed CNT films had different scattering effects on electrons and phonons due to their different mean free paths. Similar results have also been obtained in previous works (**Nat. Nanotechnol.** 2010, 5, 853; **Nat. Commun.** 2014, 5, 3848; **Adv. Funct. Mater.** 2022, 32, 2203080; **J. Mater. Chem. A** 2022, 10, 25740; **Small** 2023, 19, 2304266).”*